# Parameter optimization for MINFLUX microscopy enabled single particle tracking

Bela T. L. Vogler [1,2,3] ✉, Giovanni De Angelis[1,2], Ziliang Zhao[1,2,3], Christian Eggeling [1,2,3,4,6] ✉ & Francesco Reina [1,2,3,5,6]

MINFLUX fluorescence microscopy is a recently introduced super-resolution approach for studying cellular structures and their dynamics with highest detail. MINFLUX (*MFX*) performs Single Particle Tracking (SPT) at *runtime*. The ad hoc signal interpretation necessary to sustain the method relies on several parameters, which need to be optimized in relation to the sample under study, such as fluorescent lipid analogs in membranes, to ensure the fidelity of the measurement. We propose a parameter optimization strategy, give an overview of the most important parameters, present a theoretical upper limit for trackable diffusion rates, and demonstrate *MFX*-enabled SPT of fast ($\langle D_{\mathrm{MSD}} \rangle = 2.5 \mu m^2/s$) lateral Brownian motion of lipids in membranes.

Microscopy-based single-particle tracking (SPT) is a popular method to investigate the molecular dynamics and interactions in living systems[1–3]. SPT is supported by a large variety of microscopy techniques[2,4–7], including recently developed MINFLUX super-resolution microscopy[8,9]. MINFLUX microscopy localizes single isolated fluorescent emitters by displacing a center-symmetric excitation beam with a central intensity minimum (i.e., a *doughnut-beam*) once or multiple times in a pre-defined pattern, called the Target Coordinate Pattern (TCP), around the initial position estimate of the emitter (hereinafter referred to as a *TCP cycle or cycles*, see Methods). Imagining the discrete positions where the excitation beam is displaced to as lying on a circular path, an important parameter for further calculation is the diameter of this circle, *L* (Fig. 1a, b)[8]. The commercially available implementation of MINFLUX microscopy (*Abberior Instruments* GmbH), hereinafter referred to as MINFLUX (*MFX*), runs an iterative single particle localization routine. First, an initial position estimate is generated by searching for sources of fluorescent signal across the region of interest (Fig. 1b), which is then refined by operating successive *pattern iterations* around it at increasingly reduced *L* (Fig. 1c, see Methods). Single particle trajectories are obtained by continuously reiterating the pattern with the smallest *L*, i.e., the *last pattern iteration*, starting from the previously obtained position estimate until the signal is lost and the process restarts (Fig. 1d)[9].

In the absence of noise, the lower limit of the localization precision derived by one *pattern iteration*, when a doughnut-shaped beam is adopted for localization, is determined by the Cramer-Rao Lower Bound (CRLB) of

the maximum likelihood estimator used to locate an emitter within the TCP (see Supplementary Material of ref. 8, p. 8):

$$\sigma_{\mathrm{CRLB}}^{\mathrm{Doughnut}}(x) = \frac{L}{4\sqrt{N}} \cdot \frac{\left| \left[1 + \left(\frac{2x}{L}\right)^2\right] \cosh\left(\frac{4\ln2 \cdot xL}{v^2}\right) - \frac{x}{L}\sinh\left(\frac{4\ln2 \cdot xL}{v^2}\right) \right|}{\left|1 + \ln2 \cdot \frac{L^2}{v^2}\left[\left(\frac{2x}{L}\right)^2 - 1\right]\right|} \quad (1)$$

Here, *L* is the diameter of the TCP, *v* is the diameter (or full-width-at-half-maximum, FWHM) of the excitation beam, *N* is the number of photons collected, and *x* is the distance of the emitter to the center of the TCP.

*MFX* performs particle position estimation and trajectory linkage at runtime, i.e., during the experiment itself. Photons collected during each successful *pattern iteration* (see Methods) are evaluated on the fly to produce a particle localization, which is then used as the initial position for the subsequent *iteration*, either with a pattern of the same or decreased TCP diameter. This is in stark contrast to conventional SPT applications, where analysis of this kind usually happens in post-processing. Thus, any positional data returned by the microscope is an interpretation of the signal produced by the sample based on the given acquisition parameters like *L*, *N*, or *v*. This on-the-fly particle localization approach allows for following single emitters in time over all three dimensions of space with high spatiotemporal resolution[8,9], while delivering ready-to-analyze data. However, unifying the *ad-* and *post hoc* stages of data acquisition leads to a significant increase in process complexity and highlights

[1]Institute of Applied Optics and Biophysics, Friedrich-Schiller University Jena, Jena, Germany. [2]Department of Biophysical Imaging, Leibniz Institute of Photonic Technologies e.V., Jena, Germany. [3]Leibniz Center for Photonics in Infection Research (LPI), Jena, Germany. [4]Jena Center for Soft Matter, Jena, Germany. [5]Max Perutz Labs, Department of Structural and Computational Biology, University of Vienna, Vienna, Austria. [6]These authors jointly supervised this work: Christian Eggeling, Francesco Reina. ✉e-mail: bela.vogler@uni-jena.de; christian.eggeling@uni-jena.de

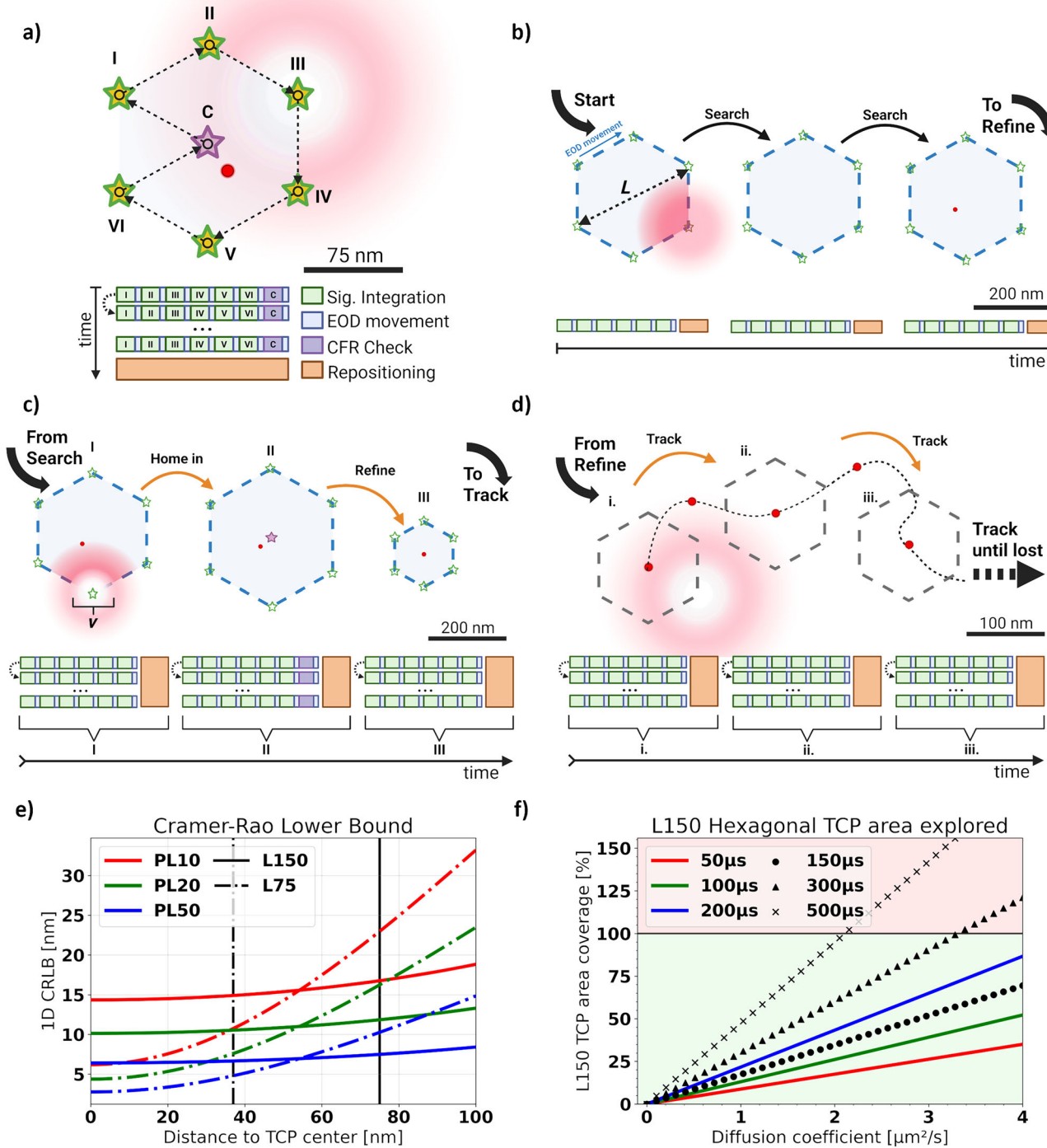

the importance of a priori considerations and choice of the *MFX* scanning parameters. In contrast to conventional SPT, maladjusted acquisition parameters may drastically limit the ability to track emitters and lead to premature interruptions when tracking a viable, i.e., not photobleached, target.

Consequently, we in the following present i) the interplay between scanning parameters (see also Supplementary Note 1) and the implications of continuous particle movement in the context of *MFX*-enabled SPT, ii) a sequence parameter optimization strategy for high sampling rate *tracking* (see also Supplementary Note 2), iii) *MFX*-enabled SPT experiments of fast lateral Brownian motion (Figs. 2 and 3), and iv) a theoretical upper limit for trackable diffusion rates (Figs. 1 and 3).

## Results and discussion
### The interplay between MINFLUX scanning parameters in the context of SPT

The *MFX* position estimation algorithm consists of two stages: (1) the initial position estimation for a detected emitter using a least mean squares estimator, and (2) the unbiasing step that corrects the initial estimate and relies on a set of pre-determined numerical coefficients[9]. Both steps are performed on the hardware provided by the manufacturer (*Abberior Instruments* GmbH). Details of the algorithm performing these operations are inaccessible to the end user. The unbiasing coefficients are provided with the instrument and are derived from the numerical solution to the problem of optimally detecting a fluorescent molecule emitting $N$ photons when

**Fig. 1 | Principle and theoretical considerations of MFX-enabled single particle tracking. a** An illustration of a single *MFX* TCP scanning pattern. The beam is steered along the marked path of vertices highlighted by green stars. The optional central scanning spot of the Center-Frequency-Ratio-Check (CFR-Check) is highlighted by a purple star. The numbers correspond to the sequence in which the scanning positions are reached. The EOD movement is represented by black arrows. The blocks on the bottom illustrate how much time it takes to execute each step of the *TCP cycle*. We represent the possibility of needing multiple *TCP cycles* in the same *iteration* to obtain a localization by repeating the block diagram along the vertical axis. In all following panels, the *Target Coordinate Pattern* (TCP) *iterations* follow this same scanning scheme. (*Created in BioRender. Vogler, B. (2025)* https://BioRender.com/f7m5i5s). **b** Illustration of the grid-search process that finds candidate emitters in the sample. We highlight how this step is performed using no beam shaping but scanning a Gaussian beam focus in a hexagonal pattern along predefined positions in the selected Region of Interest (ROI). (*Created in BioRender. Vogler, B. (2025)* https://BioRender.com/vu4km6i). **c** After a candidate emitter is detected in the search stage, the localization algorithm refines the localization estimate through successive steps in which the diameter of the TCP, *L*, is progressively reduced. The capital Roman numbers I-III correspond to the *pattern iterations* as shown in Table 1. (*Created in BioRender. Vogler, B. (2025)* https://BioRender.com/0ngjdaj). **d** Exemplification of the *MFX*-enabled Single Particle Tracking process in which the last *iteration* of the sequence (III in panel c) is repeated and the particle is followed until lost. The lower-case Roman numerals are an aid to highlight the order in which these steps are executed. In a real *MFX* experiment, the sequence will be repeated more frequently, and as such, the final *iteration* will not be repositioned by such large intervals, and this representation serves only as a visual aid. (*Created in BioRender. Vogler, B. (2025)* https://BioRender.com/yn314ae). **e** Theoretical CRLB (Eq. 1) for MINFLUX measurements executed with a doughnut-shaped beam against the distance of the localized particle to the center of the TCP for different photon limits (*PL*) and TCP diameters *L*. The vertical lines mark the TCP radius *L*/2 (solid line for $L = 150nm$, dashed line for $L = 75nm$). **f** The fraction of a $L = 150nm$ TCP area explored by a Brownian particle with constant diffusion rate *D* within different times-to-localization, i.e., the dwell time, in addition to the hardware introduced temporal overhead. The colored solid lines correspond to the dwell times used in this work. The background color corresponds to the respective area inside (green) and outside (red) of the TCP. This graph considers the time-to-localization, assuming that localizations are successfully estimated within a single *TCP cycle* ($\langle \eta \rangle = 1$).

excited by a doughnut-shaped beam scanning a TCP with a certain pattern (e.g., hexagonal) of size *L* in a certain amount of time[8–11].

While the *MFX* scanning routines (hereinafter referred to as *sequences*, see Methods), which parametrize the behavior of the microscope, contain descriptions of all *pattern iterations* of the experiment, we in this work focus on engineering only the final *iteration*, i.e., the one with the smallest *L*. The final *iteration* is the most relevant for *MFX*-enabled SPT, as its continuous reiteration realizes the tracking of single fluorescent particles. The initial *pattern iterations* are executed only once per single particle trajectory (unless otherwise specified by the user) and are used to detect emitters and to refine an initial target position instituted as a starting point for the tracking process. Nevertheless, our method is straightforwardly expanded to the remaining *pattern iterations*, as they share the same *modus operandi*. The device exports the results of all *iterations* (e.g., number of detected photons, timestamp of the localization, emitter coordinates) that produce a localization in the final dataset.

The goal of any SPT approach is to keep the time between consecutive localizations to a minimum as the particle is constantly moving, while maintaining a low level of spatial uncertainty. Between any two localizations of the particle trajectory, the *MFX* operates $\eta$ successive *TCP cycles*, each integrating fluorescent signal for a time $t_{dwell}$ (i.e., the dwell time) until a minimum number of photons, pre-defined in the sequence as the *Photon Limit* (*PL*), is acquired. Consequently, we define the time-to-localization $t_{loc} = \eta \cdot t_{dwell} + t_{hw}^{\eta}$ as the interval between any two successive localizations as determined by $\eta$, $t_{dwell}$, and the hardware overhead time $t_{hw}^{\eta}$, i.e., the delay introduced by the hardware response and execution time (e.g., beam displacement), which scales with $\eta$ and the TCP geometry (see Methods). It is important to note that because $\eta$ is dependent on the fluorescent target's Poisson statistics, $t_{loc}$ is not a single value but a digitized distribution thereof (see Methods).

Since the movement of the target diffusing particles is uninterrupted during the localization process, the capabilities of *MFX* to follow single emitters are intuitively dependent on predefined parameters like *PL*, $t_{dwell}$ and the excitation laser power (the latter determines the fluorescence emission but also photophysics such as dark time populations and photobleaching of the fluorescent label and thus the frequency of photons collected, i.e., how fast *PL* is reached)[12], the time overhead $t_{hw}^{\eta}$ and sample-dependent variables such as the signal-to-noise ratio of the collected signal. We should point out that during a localization estimation with $\eta$ cycles, photons are only integrated for the time window $\eta \cdot t_{dwell}$, while the fluorescent target is moving for the entirety of $t_{loc}$ (compare Eq. 7). This observation is, of course, valid for any and all SPT experiments; however, it is particularly significant in *MFX*-enabled SPT, since $t_{dwell}$ and $t_{hw}^{\eta}$ are in the same order of magnitude, unlike other fluorescence-based measurements.

## The implications of continuous particle movement on the MINFLUX position estimation

Assuming a Brownian particle with constant non-zero diffusion coefficient *D*, we can model the average distance traversed by the particle during a localization estimation as the standard deviation $\sigma_{Diffusion}^{d} = \sqrt{2dDt_{loc}}$ of the probability distribution of particle localizations[13], where *d* is the number of spatial dimensions and $t_{loc}$ is the time-to-localization elapsed (Eq. 7). While the position of the TCP center is updated after each successful localization, it remains unchanged during the *cycles*, i.e., during the estimation. Assuming one-dimensional diffusion for the sake of simplicity, we can estimate the average particle distance *x*, from Eq. 1 to the TCP center by the 1D standard deviation $\sigma_{Diffusion}^{d=1}$, i.e., $x = \sigma_{Diffusion}^{d=1}$. Therefore, the CRLB (Eq. 1) depends on $t_{loc}$ and on the particle diffusion rate *D*, and is thus larger for SPT when compared to imaging (i.e., localization of fixed emitters) (Fig. 1e). This difference increases when, e.g., the diameter of the TCP is reduced in an attempt to improve the localization precision without considering the system as a whole. Consequently, experimental parameters need to be chosen in accordance with each other to realize an accurate *MFX*-enabled SPT experiment: expectations of the diffusion rate, the photophysics of the fluorescent label, and relevant scanning parameters, namely *PL*, $t_{dwell}$, and *L*, must all be taken into consideration.

In the case of immobile particles ($D = 0$), this is not as problematic, as there is enough time to accurately center the TCP over the emitter, and optimization is required only to match the scanning parameter and the photophysics of the fluorescent label. This corresponds, in Fig. 1e, to the values of the CRLB where the distance from the TCP center is zero.

In summary, during position estimation, the excitation beam is successively centered on the scouting spots, i.e., the TCP vertices. Here it remains for a fraction of the dwell time during which the fluorescence signal is integrated. While this strategy is ideal for an immobile target, it poses a fundamental issue for SPT experiments. Since an *MFX* assesses the position of the tracked particle after each *pattern iteration*, given sufficient photons have been collected, it cannot refine the estimate based on the evolution of the signal during the observation. This is, however, not necessarily desirable, since such a procedure would further increase the overhead time between observations.

## Optimizing MINFLUX scanning parameters for high sampling rate SPT

As pointed out, sample-specific optimization of experimental parameters is essential to enable accurate reporting of the population of diffusing particles in *MFX*-enabled SPT measurements, which can be achieved by adjusting the TCP geometry and diameter *L*, the excitation laser power, but most importantly through *PL* and $t_{dwell}$. It is evident that minimizing the time between consecutive localizations, while keeping *PL* as high as possible, is

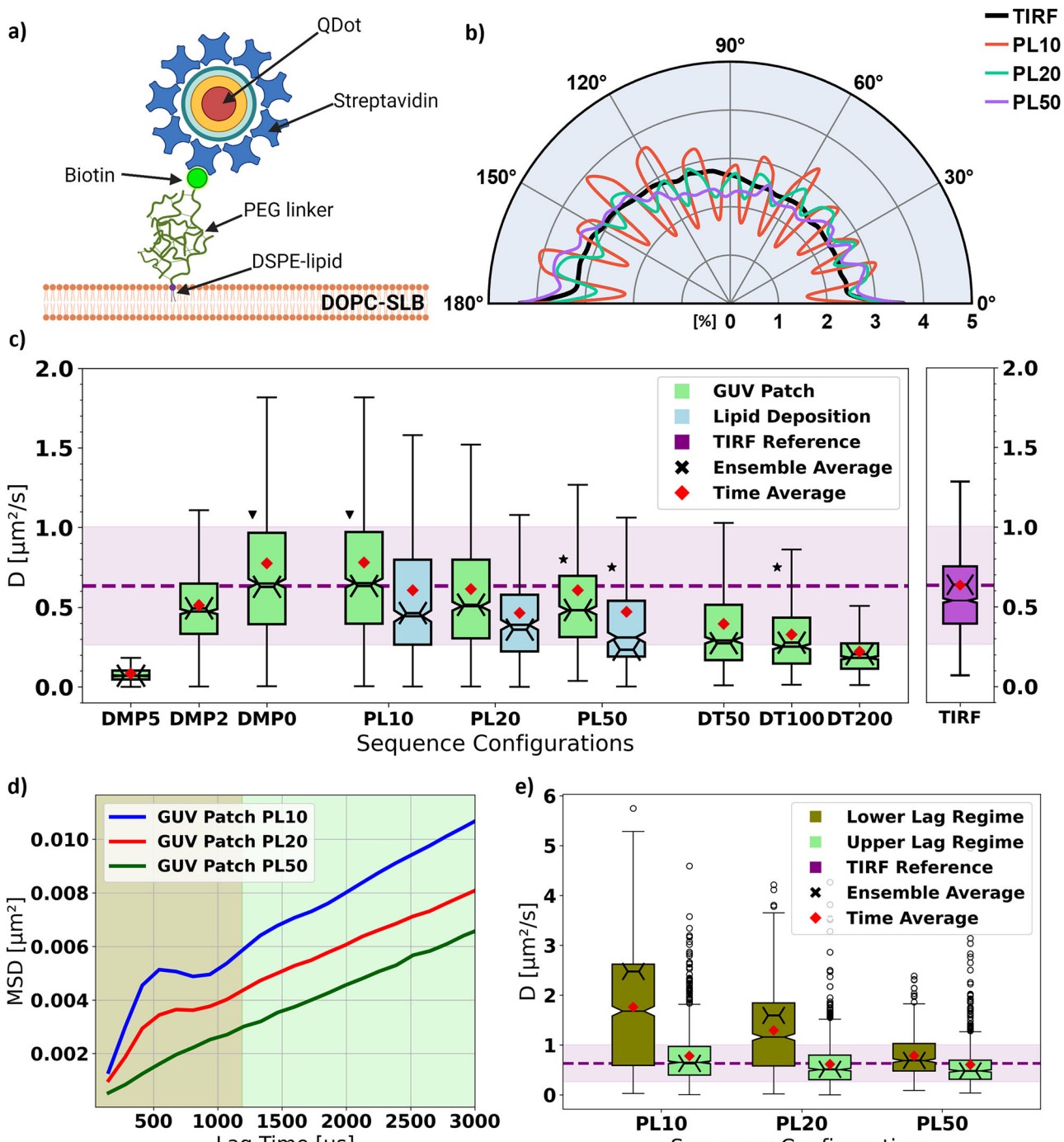

the fundamental requirement. While this premise is applicable to SPT in general, unoptimized *MFX* SPT experiments, in contrast to classical SPT, due to their ad hoc position estimation, possibly fail to return any data at all, which is in contrast to classical SPT. Given the theoretical definition of $t_{loc}$ (Eq. 5), a continuous fluorescent signal and any given *PL* and $t_{dwell}$, the time-to-localization minimizes exactly when:

$$
\begin{aligned}
1 = \langle \eta \rangle &= \left\langle \frac{N}{\psi_{\mathrm{MFX}} \cdot t_{\mathrm{dwell}}} \right\rangle \\
&= \frac{\langle N \rangle}{\langle \psi_{\mathrm{MFX}} \rangle \cdot t_{\mathrm{dwell}}} = \frac{\langle N \rangle}{\langle \psi_{\mathrm{MFX}} \cdot t_{\mathrm{signal}}^{\eta=1} \rangle} \geq \frac{N_{\mathrm{PL}}}{\langle \psi_{\mathrm{MFX}} \cdot t_{\mathrm{signal}}^{\eta=1} \rangle}
\end{aligned}
\tag{2}
$$

where $\langle \psi_{\mathrm{MFX}} \rangle$ is the average photon detection frequency per localization (sometimes referred to as EFO), $\langle N \rangle \left( = \langle \psi_{\mathrm{MFX}} \cdot t_{\mathrm{signal}}^{\eta} \rangle = \langle \psi_{\mathrm{MFX}} \cdot \eta \cdot t_{\mathrm{dwell}} \rangle = \langle \psi_{\mathrm{MFX}} \rangle \cdot \langle \eta \rangle \cdot t_{\mathrm{dwell}} \right)$ the average number of collected photons (sometimes referred to as ECO), $t_{\mathrm{dwell}}$ the dwell time per localization round as put in the sequence, and $N_{PL}$ the photon limit.

Consequently, to enable reliable position estimation, *PL* should be matched with the expected average photon count $\langle N \rangle$ per unit time $t_{\mathrm{signal}}^{\eta=1}$. It is even desirable to set the *PL* to a slightly underestimated value for the average emission of the target fluorophore. This will ensure that, on average, each *TCP cycle* will lead to a successful localization, thereby maximizing the sampling efficiency in terms of minimizing the time-to-localization.

**Fig. 2 | Experimental MFX-enabled single particle tracking (SPT) of QD-labeled lipid on a supported lipid bilayer (SLB)—dependencies on hardware parameters. a** Biotinylated lipid analogs (DSPE-PEG-biotin, see Methods) tagged with streptavidin-coated metallic core quantum dots (QDs) were incorporated in homogeneous DOPC SLBs and diffused freely in the membrane. (*Created in BioRender. Vogler, B. (2025)* https://BioRender.com/vh9i314). **b** Spline interpolated distribution of turning angles between successive displacement vectors of different ensembles of lipid trajectories, detected using *MFX*-SPT with different photon limits (PL) and the same TCP diameter ($L = 150$ nm). As a reference, we reported the same distribution obtained using SPT experiments performed on a custom TIRF setup. **c** Comparison of the measured diffusion coefficients across different sequence configurations tested on SLBs generated using GUV patches (green) or flow chamber lipid deposition (blue): different damping parameter *DMP* (with $PL = 10$ and $dT = 100\mu s$), photon limit *PL* (with $dT = 100\mu s$ and $DMP = 0$), and dwell time $t_{dwell}$ (with $PL = 50$ and $DMP = 0$) settings. Each boxplot (center line, median; red diamond, mean; box limits, first to third quartile; whiskers, 1.5x interquartile range; notch, height proportionally to interquartile range) corresponds to the distribution of diffusion coefficients extracted from the detected trajectories using the OLSF routine on the *Upper* Lag Regime of the MSD curves. We highlight the time-average (red diamond) and ensemble-average (black cross) diffusion coefficients for each dataset. As a reference, we provide the distribution of diffusion coefficients (average diffusion coefficient (purple line) and standard deviation (lilac-shaded area)

determined from the TIRF experimental data on a QD labeled GUV-patch SLB using the same analysis routine. The respective distribution of diffusion coefficients can be found as a boxplot (center line, median; red diamond, mean; box limits, first to third quartile; whiskers, 1.5x interquartile range; notch, height proportionally to interquartile range) to the right. Given the equality of time-average and ensemble-average diffusion coefficient for this dataset, we chose to display only one line. Each triplet of datasets focusing on a specific parameter, e.g., *PL*, has been taken on separate samples, with the exception of the GUV Patch *DMP* and *PL* sets, which share the sample. Two datasets have been taken on the same sample with the same sequence ($PL = 10$, $DMP = 0$, $dT = 100\mu s$; black triangle), while three other sets have been taken on different samples with the same parameters ($PL = 50$, $DMP = 0$, $dT = 100\mu s$; black star). **d** Illustration of how different values of *PL* contribute to generating the two Lag Regimes observed in our SPT experiments using the ensemble average MSD curves calculated from our GUV-patch SLB experiments. The shading highlights the two regimes (Lower Lag Regime: olive, Upper Lag Regime: light-green). We use an arbitrary threshold of 9 lags ($\approx 1190\mu s$) to separate the regimes. **e** Comparison boxplot (center line, median; red diamond, mean; box limits, first to third quartile; whiskers, 1.5x interquartile range; notch, height proportionally to interquartile range; points, fliers) of the extracted diffusion coefficients (as in (**c**)) for the GUV-patch SLB experiments between the Lower and Upper Lag Regimes, together with the reference values from the TIRF experiments (as in (**c**)).

However, if the photon limit is set too low, the localization algorithm does not take full advantage of the photon budget afforded by the target fluorophore. As pointed out before, while a slightly higher number of detected photons compared to *PL* can still prove beneficial, significantly larger numbers will cause positional offsets during the unbiasing step, which follows position estimation[14]. These errors become even more severe for larger $L$, as the approximation of the central minimum of the excitation beam (i.e., the *doughnut-beam*) by an ideal parabola (as done in practice) does not hold in that case, causing a larger bias in the estimator.

### Determination of a parametric expression for the upper limit of trackable diffusion rates in two-dimensional MINFLUX enabled SPT

Given that emitters are only tracked while inside the TCP, there is a risk for them to escape should the area $\pi\left(\sigma_{Diffusion}^{d=1}\right)^2$, which is explored by an emitter during the time-to-localization $t_{loc}$, approach the TCP area $\pi(L/2)^2$. This solidifies an upper limit for trackable diffusion rates $D$ given a specific $t_{loc}$ (Fig. 1f). If a particle moves with a diffusion rate beyond this limit, *MFX* will not be able to produce reliable localizations, if any at all. From Eq. 1 it becomes clear that while the spatial resolution is increased linearly by reducing $L$, the risk of losing the emitter rises quadratically.

For simplification, we may set the Brownian diffusing emitters to exhibit a constant average emission rate $\langle\theta_P\rangle$ during an optimized photon acquisition ($\langle\eta\rangle = 1$) and assume an approximately constant average photon detection rate $\langle\psi_{MFX}\rangle \leq \langle\theta_P\rangle$. In this case, we can give an estimation of the upper limit of possibly trackable two-dimensional (2D) diffusion rates $D_{max}^{2D}$ under the condition that the particle must remain within the area of the TCP during the localization process (see Methods).

$$D_{max}^{2D} = \frac{L^2}{16} \cdot \frac{\langle\theta_P\rangle}{N_{PL} + \langle\theta_P\rangle t_{hw}^{\eta=1}} \leq \frac{L^2}{16} \cdot \frac{\langle\psi_{MFX}\rangle}{N_{PL} + \langle\psi_{MFX}\rangle t_{hw}^{\eta=1}} \quad (3)$$

### Experimental observation of the influence of MINFLUX scanning parameters

From our theoretical considerations, we know that the excitation laser power, *PL*, and $t_{dwell}$ are the most impactful parameters to consider when optimizing the *MFX* time-to-localization (see Methods). While the choice of

laser power strongly depends on the sample used (e.g., the brightness or photobleaching properties of the target fluorophore,) both *PL* and $t_{dwell}$ can be chosen arbitrarily before the experiment. Consequently, we experimentally investigated the influence of these two parameters on *MFX*-enabled SPT measurements. To this end, we performed SPT of biotinylated lipid analogs (DSPE-PEG-Biotin, tagged with streptavidin-coated Quantum Dots (QDs)) embedded in a homogeneous fluid continuous Supported Lipid Bilayer (SLB, unsaturated 1,2-Dioleoyl-sn-glycero-3-PC (DOPC) lipid, see Methods)[15] (Fig. 2a). We intentionally employed large ($\approx 15nm - 20nm$ in diameter[16]) and bright metallic core QDs as luminescent tags to slow down the diffusion of the target biotinylated lipid analogs while preserving their characteristic free diffusion[17–20]. The employed SLBs were generated either by lipid depositions in enclosed flow-chamber systems or by Giant Unilamellar Vesicle (GUV) patching. We performed the main part of our experiments on GUV patches, as they offer an easy and well-explored model system.

Table 1 lists all relevant experimental *MFX* sequence parameters as implemented in *MINFLUX-IMSPECTOR* (commercial version-16.3.15645-m2205). With $L = 150nm$, Stickiness = 4 and MaxOffTime = $3ms$ (see Supplementary Note 1, Parameter Overview), in our experiments, we chose to stay lenient during tracking to catch as many datapoints as possible, which gave us more freedom in data analysis but required *post hoc* artifact removal.

In our analysis, we split trajectories whenever the number of *cycles* $\eta$ between consecutive localizations exceeded $\delta = \langle\eta\rangle + \text{std}(\eta)$, which effectively removed large time gaps, mid-trace particle-swap events, and jitter. Further, using our previous SPT analysis pipeline, we eliminated artifacts due to, e.g., label-induced cross-linking of various lipids[21]. We then finally extracted the lateral diffusion coefficient $D_{MSD}$ and the dynamic localization uncertainty $\sigma_{MSD}$ from a custom Mean Squared Displacement (MSD) implementation suited for inhomogeneous time-lags using Optimal Least Squares Fitting (OLSF)[22]. As a reference for the obtained values of $D_{MSD}$, we performed SPT on the same QD-labeled lipids in a similar SLB using a custom-built microscope with Total-Internal-Reflection Fluorescence (TIRF) excitation and camera-based detection[23], and analyzed the data using the same pipeline (Fig. 2c right side).

Before detailing any biasing influence of *PL* and $t_{dwell}$ on the experimental diffusion data, we first inspected another experimental *MFX* parameter, the galvo damping *DMP* (see Supplementary Note 1, Parameter Overview). We reviewed values of $D_{MSD}$ obtained for different *DMP* (Fig. 2c, left side, *PL* set to 10). Since $DMP > 0$ led to an underestimation of particle

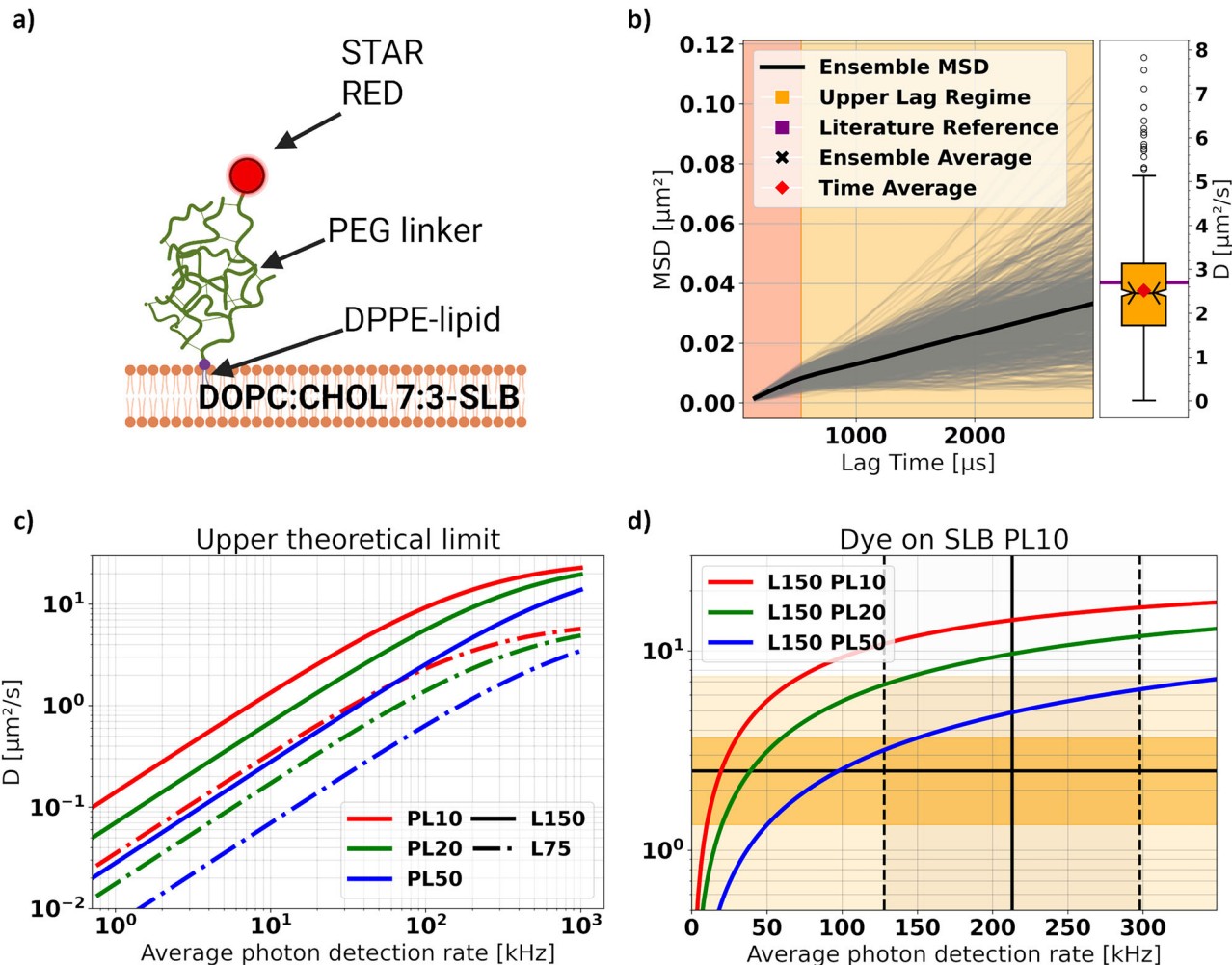

**Fig. 3 | Theoretical and experimental considerations reveal the upper limit of trackable diffusion rates. a** Fluorescent lipid analogs (DPPE-PEG2000-STAR RED) tagged with Abberior STAR RED were incorporated in homogeneous DOPC:CHOL 7:3 bilayers. (Created in BioRender. Vogler, B. (2025) https://BioRender.com/tmaaqpx). **b** Single-molecule MSD curves (gray lines), and the ensemble average MSD curve (black line) calculated for trajectories of the fluorescent lipid analogs diffusing in the SLB illustrated in (**a**). The shading corresponds to the Lower (orange-red) and Upper (orange) Lag Regimes, with an arbitrary threshold set at 5 lags ($\approx 670\mu s$). The corresponding distribution of diffusion coefficients (time-average (red diamond) and ensemble-average (black cross)) extracted for the Upper Lag Regime of the MSD can be found as a boxplot (center line, median; red diamond, mean; box limits, first to third quartile; whiskers, 1.5x interquartile range; notch, height proportionally to interquartile range; points, fliers) to the right. The

horizontal purple line marks a system-specific reference value taken from literature[27]. **c** Upper theoretical limit of trackable diffusion coefficients $D$ against the average photon detection rate of the particle within one *pattern iteration* as calculated from Eq. 3, in the absence of noise. Different line styles and colors represent unique TCP diameters $L$ and photon limits PL as labeled. **d** Zoom-in of the graph in c) for values of $D$ and average photon detection rate more common in experimental practice. The black horizontal line indicates the ensemble (and time) average diffusion coefficient $D$ for the dataset in panel b), while the vertical one represents the average frequency of photon detection $\langle \psi_{iMFX} \rangle$. The shaded dark orange area represents the standard deviation of the distribution of $D$ from panel b), whereas the lighter orange represents the full extent of the distribution. The vertical dashed lines represent the standard deviation of the photon detection rate.

---

motility, we generally kept the galvo damping off ($DMP = 0$) in our experiments.

Turning now to PL and $t_{dwell}$, we first tested possible bias in the *MFX* data by checking whether the tracks showed an isotropic progression as expected for Brownian diffusion. The Turning Angle Distribution (TAD) demonstrates an additional geometric bias in that a set of directions is preferred with decreasing PL (Fig. 2b, $dT = 100\mu s$), digitizing the direction of positional updates, i.e., too low values of PL resulted in a biased detection of diffusion. This is a direct result of the hexagonal TCP geometry in combination with the reduced number of acquired photons and the thus reduced size of the statistical data sample available for analysis (compare Fig. 2b–d in ref. 9).

As mentioned above, our isotropic progression analysis highlighted a favorable use of larger PL values. Unfortunately, we had to reduce PL to avoid missing faster diffusion events, i.e., to sample a higher bandwidth of

particle diffusion rates, highlighted by an artefactual dependency of the extracted values of $D_{MSD}$ on PL (Fig. 2c). While higher PL entail a smaller localization uncertainty $\sigma_{MSD}$, i.e., an in principle higher spatial precision (Table 2), the determination of values of $D_{MSD}$ became biased towards slower moving emitters, i.e., emitters moving too fast were lost and the reported values of $D_{MSD}$ appeared to underestimate the reference value of the TIRF-SPT experiments (Fig. 2c and Table 2). We observed the same characteristics for both kinds of SLB sample preparation. The coinciding change of $\sigma_{MSD}$, $D_{MSD}$, and PL can be traced back to the CRLB's dependence on the number of photons acquired and the aforementioned increased possibility of losing particles moving too fast for too long times ($t_{loc}$), i.e., the time required in total for a localization (Fig. 1e, f).

We next investigated the effect of different $t_{dwell}$ on the experimentally detected diffusion rates, while maintaining PL = 50 (Fig. 2c). Confirming Eq. 5, we revealed larger average values of $D_{MSD}$ for smaller $t_{dwell}$, reaching

**Table 1 | Experimental MINFLUX sequence parameters for 2D single particle tracking of fluorescent dye or QDot-labeled lipid analogs in the SLB**

| 2D Tracking | Pinhole orbit [I] | 1st Pattern iteration [II] | 2nd Pattern iteration [III] |
|---|---|---|---|
| $L$ (nm) | 284 | 302 | 150 |
| Pattern shape | Hexagon | Hexagon | Hexagon |
| Photon limit (counts) | 40 | 20 | 50 (*) |
| Laser power factor (times) | 1.0 | 1.5 | 2.0 (*) |
| Pattern dwell time (µs) | 500 | 100 | 200 (*) |
| Pattern repeat (times) | 1 | 1 | 1 |
| CFR threshold | −1.0 | 0.5 | −1.0 |
| Background threshold (kHz) | 15 | 30 | 30 (*) |
| MaxOffTime (control param.) | 3 ms | 3 ms | 3 ms |
| Damping (control param.) | 0 | 0 | 0 |
| Headstart (control param.) | −1 | −1 | −1 |
| Stickiness (control param.) | 4 | 4 | 4 |

(*) marks the parameters that were adjusted during the experiments.

the reference value of the TIRF-SPT experiments only for very small $dT \le 50\mu s$. Yet, it must be kept in mind that due to the additional hardware-induced overhead time $t_{hw}^{\eta}$, which is in the same order of magnitude as the dwell time $t_{dwell}$, the time-to-localization $t_{loc}$ has a lower limit. Given that photon emission follows Poisson statistics and assuming a fixed average emission rate $\langle \theta_p \rangle$, the average number of detected photons per *cycle* $\langle N_P^{\eta} \rangle$ is linearly dependent on the time window of signal integration ($t_{signal}^{\eta} = \eta \cdot t_{dwell}$). Therefore, for lower $t_{dwell}$ a larger average number of *TCP cycles* $\langle \eta \rangle$ is required to collect enough photons (*PL*) per successful localization (compare Table 2). This again results in a larger or equal $t_{loc}$ (compare Eq. 4) given that each *TCP cycle* causes additional hardware overhead. As a reminder, $t_{loc}$ is determined by the number of *TCP cycles* $\eta$ times the dwell time $t_{dwell}$ plus the overhead time $t_{hw}^{\eta}$ (Eq. 7).

In a nutshell, for a given $\langle \theta_P \rangle$, fewer and shorter *cycles* per localization mean a higher frequency of position updates, which permits following faster particles at a reduced distance from the TCP center, minimizing the localization error induced by the particle movement, i.e., the offset between the actual (and unknown) particle position and the estimation produced. This strategy effectively improves the overall spatio-temporal resolution.

Improving the localization precision by collecting more photons over a longer integration time ($t_{signal}^{\eta}$) may not always be viable, since increasing $t_{loc}$ in turn increases the deviation between estimation and actual particle position or renders the position estimation impossible altogether. This is due to an increase in the area explored by the diffusing particle during the photon collection (compare Fig. 1f). To counteract this effect, it may appear reasonable to adopt TCP scanning patterns with fewer positions to increase the sampling rate. This has already been employed in the past[8,9], and it does marginally increase the temporal resolution, since fewer positions have to be scanned during one TCP, thus shortening $t_{hw}^{\eta}$. However, reducing the number of vertices in the position estimator also results in an undesirable increased angular bias, which is especially true for low numbers of photons (Fig. 2b). While in principle, any TCP with a finite number of vertices (i.e., not an ideal circle) will result in some degree of angular bias, the hexagonal geometry has proven to be a suitable compromise between temporal efficiency and angular resolution (compare Fig. 2b–d in [9]). Therefore, adopting a hexagonal TCP is preferable, at least when probing diffusive motion with a high degree of randomness[8].

While calculating the ensemble-MSD from our experiments, we noticed that they expressed two discontinuous linear regimes, most prominently for low values of *PL*, i.e., high temporal resolutions (Fig. 2d). Though varying in severity, we found the *kink*-like shape in all of our fastest datasets, i.e., the datasets of highest temporal resolution. Consequently, for the entirety of this work, we distinguish between the *Lower* Regime MSD, meaning the part with or prior to the *kink*-like shape, and the *Upper* Regime, referring to the continuous linear part beyond that. Here, the words *Lower* and *Upper* refer to the interval of time lags $\tau$ in which discontinuity appears. We noticed a correlation between the severity of the *kink* and the employed *PL* (Fig. 2d). Given that a lower *PL* enables faster sampling (see Methods and Table 2), we further noted a correlation between the sampling frequency of *MFX* ($\sim 1/\langle \eta \rangle$) and the severity of the discontinuity of the MSD curve. The same was true for the dynamic localization error $\sigma_{MSD}$, which got worse (i.e., increased) with increased sampling frequency (Table 2).

We excluded the possibility of strong confined or compartmentalized diffusion causing these two-regime characteristics in our experimental MSD, since our membrane bilayers were very fluid and homogeneous, consisting of one type of unsaturated lipid (see Methods). Therefore, and according to previous work[24], it is most probable that the *Lower* Regime MSD did not result from the diffusive motion of the lipid but was rather due to the wiggling motion of the QDs ($\approx 15nm - 20nm$ in diameter[16]) attached to the PEG(2000) linkers ($\approx 3nm$ in diameter[25]) on top of the particle (compare Fig. 5 in ref. 24). This became apparent when comparing the extracted values of $D_{MSD}$ for both *Lower* and *Upper* Regime (Fig. 2e). On one hand, the *Upper* Regime exhibited diffusional rates that were close to those of our TIRF microscopy reference experiments and marginally dependent on *PL*, highlighting the true mobility of the lipid. On the other hand, $D_{MSD}$ values of the *Lower* Regime were far above those of the references (Fig. 2e). Therefore, we considered the *Upper* Regime as our region of interest. Throughout this work, except for Fig. 2e, we list only the diffusion rates extracted from the *Upper* MSD Regime (Figs. 2c and 3b; Table 2). Further investigations into the cause of the MSD discontinuity in general were beyond the scope of this work. Nevertheless, we should point out that this phenomenon must be taken into consideration in the larger context of SPT, given that a significant quantity of analysis approaches in the literature merely consider the first few points in the MSD curves to estimate the diffusion rate of the targets of interest[26].

Given that Brownian motion is a homogeneous, isotropic, and memoryless process, particle systems exhibiting Brownian motion are expected to be ergodic, i.e., to have an ergodic ratio $\varepsilon_{Brownian} = 1$. Therefore, it is important to point out that all MINFLUX datasets, which reported diffusion coefficients close to the values of the TIRF microscopy reference, were characterized by an ergodic ratio of about $\varepsilon_{QD} \approx 0.8$ as compared to $\varepsilon_{TIRF} \approx 1.0$ for TIRF. Following the above discussion, the observed ergodicity break was most likely due to the superposition of the wiggling motion of the QDs and the particle diffusion in the SLB.

## Accurate tracking of fast fluorescent lipid analogs on model membranes with optimized MINFLUX enabled SPT

Next, we conducted *MFX*-enabled SPT experiments on fluorescent lipid analogs (Dipalmitoylphosphatidylcholine tagged with Abberior STAR-Red via a PEG-linker, DPPE-PEG) embedded within a GUV-patch derived SLB (70% DOPC and 30% cholesterol (CHOL), see Methods; Fig. 3a). Compared to QDot-labeled lipids, these fluorescent lipid analogs have much higher diffusion rates and should approach the limit of what is feasibly trackable by the device. The addition of cholesterol was made to marginally slow down the diffusion (to ensure staying within $D_{max}^{2D}$) while maintaining its characteristic isotropic and unconfined movement. Following the same routine as before (with $PL = 10$, $dT = 100\mu s$, $DMP = 0$), we calculated the MSD. From its *Upper* Regime, we obtained an ensemble and time-average diffusion coefficient $D_{Dye} \approx 2.5\mu m^2/s$ and an ergodicity coefficient of $\varepsilon_{Dye} \approx 0.98$, i.e., close to perfect homogeneous diffusion (Fig. 3b and Table 2).

**Table 2 | Experimental results reveal an optimization tradeoff between trackable diffusion rates and localization error for MFX-enabled single particle tracking**

| S | PL | DMP | $t_{dwell}$ | $D_{eMSD}$ [μm²/s] | $σ_{eMSD}$ [nm] | $σ_{CRLB}$ [nm] | $⟨η⟩$ | $⟨t_{loc}⟩$ [μs] | $\frac{(N)}{⟨η⟩N_{PL}}$ | n |
|---|----|-----|-------------|--------------------|------------------|------------------|-------|-------------------|--------------------------|---|
| I | 10 | 0 | 100 | 2.46 | 30.30 | 20.03 | 1 | 150 | 2.13 | 855 |
| II | 10 | 5 | 100 | 0.07 | 16.52 | 20.03 | 1 | 150 | 4.10 | 924 |
| II | 10 | 2 | 100 | 0.49 | 15.53 | 20.03 | 1 | 150 | 4.49 | 988 |
| II | 10 | 0 | 100 | 0.63 | 28.02 | 20.03 | 1 | 150 | 3.36 | 1075 |
| II | 10 | 0 | 100 | 0.63 | 27.96 | 20.03 | 1 | 150 | 3.36 | 1061 |
| II | 20 | 0 | 100 | 0.52 | 22.78 | 14.35 | 1 | 151 | 2.27 | 841 |
| II | 50 | 0 | 100 | 0.48 | 14.05 | 9.08 | 2 | 282 | 0.71 | 534 |
| III | 10 | 0 | 100 | 0.46 | 30.37 | 20.03 | 1 | 150 | 1.53 | 576 |
| III | 20 | 0 | 100 | 0.36 | 23.51 | 14.35 | 2 | 281 | 0.67 | 325 |
| III | 50 | 0 | 100 | 0.23 | 13.46 | 9.08 | 5 | 676 | 0.22 | 323 |
| II | 50 | 0 | 50 | 0.28 | 4.67 | 9.08 | 5 | 437 | 0.23 | 779 |
| II | 50 | 0 | 100 | 0.28 | 5.50 | 9.08 | 3 | 414 | 0.42 | 763 |
| II | 50 | 0 | 200 | 0.21 | 10.91 | 9.08 | 2 | 483 | 0.82 | 564 |
| II | TIRF SPT | | | 0.64 | 39.86 | - | 1 | 14925 | - | 1810 |

In addition to the ideal CRLB $σ_{CRLB}(\bar{r} = \bar{0})$ (see Supplementary Material of ref. 8, p. 11) we list the lateral diffusion coefficient $D_{eMSD}$ as well as the lateral localization error $σ_{eMSD}$ extracted from the ensemble-average Mean-Squared-Displacement (eMSD) using Optimal Least Squares Fitting (OLSF) assuming momentary Brownian motion. Here $η$ refers to the number of cycles, $t_{loc}$ is the time-to-localization, $N$ is the number of photons, $N_{PL}$ is the PL, and $⟨...⟩$ the scale-appropriate average value. We indicate the sample size $n$, i.e., the number of tracks per set, in the last column. The first column $S$ indicates the sample in reference: Dye on GUV-patch SLB (I), QDs on GUV-patch SLB (II), QDs on Lipid-Deposition-SLB (III).

In comparison to the QD-tagged samples, we noticed that the kink-like shape of the ensemble-MSD, though retained, was significantly more subtle. This implies reduced wobbling of the PEG-linker tagged with the dye rather than with the QDs, which was reflected in the contrast of ergodicity coefficients $ε_{QD} ≈ 0.8$ and $ε_{Dye} ≈ 1.0$ for the same sequence parameters (i.e., $PL = 10$, $dT = 100μs$, $DMP = 0$). Consequently, MFX successfully observed the expected isotropic Brownian Motion with a diffusion rate close to values reported in relevant literature[27].

We then compared our experimental results to theory. Figure 3c plots diffusion rate limits $D_{max}^{2D}$ (Eq. 3) for different values of $L$, $PL$ and $⟨ψ_{MFX}⟩$, highlighting the best chances to track fast-moving emitters for large $L$ (e.g., $150nm$) and $⟨ψ_{MFX}⟩$ and low $PL$ (e.g., 10). With photon emission frequencies in the range of $125kHz − 300kHz$ (as in our experiments), MFX should then be able to track emitters moving as fast as $D_{max}^{2D} ≈ 10μm²/s − 11μm²/s$ with optimized parameters (Fig. 3d). Our experimental data of Fig. 3b ($D_{Dye} ≈ 2.5μm²/s$, taken with $PL = 10$ and $t_{dwell} = 100μs$) was thus well within these theoretical boundaries (Fig. 3d). With these experimental settings, slightly more than $N = 20$ photons were detected on average per localization and cycle ($PL = 10$, first row in Table 2), which resulted in an average photon detection frequency per localization $⟨ψ_{MFX}⟩ = ⟨N/ηt_{dwell}⟩$ of around 213 kHz ($⟨t_{loc}⟩ = 150μs$, Table 2 and Eq. 3). One might argue that increasing the $PL$ to 20 could prove beneficial in terms of localization precision while maintaining an accurate value for the average diffusion rate. Further increase of the $PL$ or $t_{dwell}$, or decreasing $L$ for that matter, would, however, result in an underestimation of the particle motility. It is important to note that we always assume $⟨η⟩ = 1$, which implies $t_{dwell} \sim 1/⟨ψ_{MFX}⟩$ and $N_{PL} \sim ⟨ψ_{MFX}⟩$ (compare Eq. 3).

## Conclusions

We theoretically and experimentally demonstrated the implications of the iterative localization procedure of MFX-enabled SPT and highlighted the resulting critical dependence on pre-defined acquisition parameters such as $L$, $PL$, and $t_{dwell}$. We deducted an upper limit of reliably trackable diffusion rates, introduced a parameter optimization scheme (Supplementary Note 2, MINFLUX Sequence Optimization Guide for 2D SPT Experiments) to allow for the capture of a broader bandwidth of fast diffusion events, and provided an overview of the most relevant MFX sequence parameters (Supplementary Note 1, Overview of the Most Influential Sequence Parameters in MINFLUX enabled Single Particle Tracking) alongside practical tips and warnings for MFX sequence manipulation. Finally, we experimentally demonstrated the possibility of reliable MFX-enabled SPT of fast ($⟨D_{MSD}⟩ = 2.5μm²/s$) lateral Brownian diffusion of fluorescent lipid analogs in model membranes.

So far, we have restricted ourselves to two-dimensional experiments. However, our discussion and findings can be easily extended to the three-dimensional case, given that they share the same modus operandi. We stress that all the considerations for Brownian diffusing particles in this text also apply to cases of directed or processive motion, such as those considered in refs. 28,29, since the instrument adopts the same detection strategy irrespective of the experiment.

## Methods

### Continuous supported lipid bilayer composition and formation

The continuous supported lipid bilayer (SLB) used as a target sample was composed of DOPC (1,2-dioleoyl-sn-glycero-3-phosphocholine, Avanti Polar Lipids, Inc.) with the addition of 0.01Mol% of the fluorescent lipid analog DOPE-ATTO488 (ATTO-TEC GmbH) for ease of identification under confocal fluorescence imaging. The targets of SPT were DSPE (1,2-distearoyl-sn-glycero-3-phosphoethanolamine) lipid analogs with a PEG(2000)-Biotin linker (Avanti Polar Lipids, Inc.), added in $0.15 \cdot 10^{-3}$Mol% to the lipid solution above. The SPT targets were labeled with streptavidin-coated metallic core QDs, whose fluorescence emission spectrum manifests as a symmetric peak centered around 655 nm (QDs 605, ThermoFisher, Inc.).

The continuous SLB production was based on a previously reported solvent-assisted lipid bilayer (SALB) formation method[15]. Lipids were first dissolved in chloroform upon arrival and kept as stock solutions at $−20\,°C$. Before use, the desired amount of lipid stock solution was blow dried by a gentle stream of nitrogen gas and the lipid film was resuspended in isopropanol (IPA) at a concentration of $0.5mg/ml$. Glass coverslips (Epredia, $0.16 − 0.19mm$, $26 × 76mm$) were rinsed sequentially with ethanol, ultrapure water and cleaned with detergent, they were treated with a plasma cleaner (Zepto One from Diener Electronic GmbH, Plasma-Surface-Technology) for 1 min before the assembly of the microfluidic flow channel with Ibidi GmbH sticky-Slide I Luer (0.1mm channel height), the flow

channel was later connected to a *Hamilton, Inc.* gas tight glass syringe (2.5*ml*) via appropriate connectors and tubing. A high-precision syringe pump (*CETONI GmbH* Nemesys S) was used to control the liquid exchange. The flow channel was initially filled with PBS buffer, then the buffer was replaced with IPA at a flow rate of 50*μl*/ min for 5 min, target lipid/lipid mixtures in IPA was then introduced into the channel at 50*μl*/ min for 2 min and with incubation on the coverslip for another 5 − 10 min, finally PBS buffer was again introduced into the flow channel at 50*μl*/ min for 2 min for complete formation of continuous SLB, and with a subsequent increased flow rate at 100*μl*/ min for 1 min to rinse off loosely attached lipid structures.

### Giant unilamellar vesicle patch SLB composition and formation
The lipids used for these samples are DOPC (1,2-dioleoyl-sn-glycero-3-phosphocholine, *Avanti Polar Lipids, Inc.*), Cholesterol (Ovine, cholest-5-en-3β-ol, *Avanti Polar lipids, Inc.*), Atto 488-DOPE (1,2-Dioleoyl-sn-gly-cero-3-phosphoethanolamine, Atto-Tec), DSPE-peg2000-biotin (1,2-di-stearoyl-sn-glycero-3-phosphoethanolamine-N-[biotinyl(polyethylenegly-col)-2000] (ammonium salt), *Avanti Polar Lipids, Inc.*) and DPPE-peg2000-STAR RED (1,2-dipalmitoyl-sn-glycero-3-phosphoethanolamine-N-[azi-do(polyethyleneglycol)-2000, Abberior).

We prepared Giant Unilamellar Vesicles (GUVs) through electroformation similar to refs. 23,30 in sucrose solution (300*mM*), using a solution of DOPC and DOPC:CHOL 7:3, depending on the experiment shown. In both cases, we added DOPE Atto 488 (0.01Mol%). In the experiments in which QDs are used, we added to the lipid solution also DSPE-PEG2000-Biotin (0.01Mol%). We plasma cleaned the coverslip to rupture the GUVs and create GUV patches, and we used, as buffer, phosphate buffer saline (PBS, 137*mM* NaCl, 10*mM* phosphate, 2.7*mM* KCl) to keep the supported lipid bilayer (SLB) hydrated. We then labeled the biotinylated lipids with streptavidin conjugated 655 QDs (1*μM* concentration, Invitrogen by Thermo Fisher Scientific) at a concentration of 50*pM* for MINFLUX SPT, while we used streptavidin conjugated 605 QDs (1*μM* concentration, Invitrogen by Thermo Fisher Scientific) at a concentration of 1*pM* for TIRF single particle tracking. In the experiments in which we track the lipid analog DPPE-peg2000-STAR RED, we labeled the SLB with a concentration of 50*pM*.

### TIRF data acquisition
The TIRF single particle tracking data are acquired on a custom-made iSCAT-TIRF microscope as described in ref. 23. We excited the 605 QDs with a 488 nm diode laser and, to efficiently detect the fluorescence signal, the filters in the TIRF's imaging channel have been changed accordingly. We acquired 3000 frames for each measurement with an exposure time of 10*ms* that led to an effective frame time of 67*ms*.

### MINFLUX single particle tracking
The SPT experiments were performed on a commercial MINFLUX microscope setup (*Abberior Instruments* GmbH), which is based on an iterative localization approach[9]. For this study, we restricted our investigation to 2D tracking, although the setup is capable of 3D tracking. The device was provided with a set of default localization routines for different purposes, from SPT to imaging, for 2D and 3D experiments, where a large number of parameters can be arbitrarily modified and adjusted by the users, within limits and a procedure specified by the manufacturer. The scanning parameters relevant for our experiments are reported in Table 1. We note that the pattern dwell time ($t_{dwell}$), i.e., the total time that the MINFLUX setup spends collecting photons during one TCP *cycle*, could be altered without much consequence. On the other hand, the photon limit (*PL*) and TCP diameter (*L*) required additional changes to the parameters provided for the localization correction, as the CRLB (Eq. 1) is dependent on both. They are available at "…/seq/containers/*.json". We highlight that another variable that changed between *pattern iterations* was the excitation laser power. We express this by listing amongst the parameters a laser power

multiplier, which refers to a starting excitation power of 10*μW* at the sample plane.

### MINFLUX localization estimation
MINFLUX realizes processive particle localization by employing a chain of successive *pattern iterations*. Behavior during these is governed by the *MFX-*sequence parameters (Supplementary Note 1, Parameter Overview). Each *iteration* follows the same core principles (Fig. 1a).

Around an initial assumed particle localization, the scanning beam is steered along a pre-defined path called the Target Coordinate Pattern (TCP) with its vertices on a virtual circle of diameter *L* using Electro Optical Deflectors (EODs). At each vertex, the signal is integrated for a time ($t_{integration}$) equal to $t_{integration} = t_{dwell}/(N_{vertices} \cdot N_{TCP})$, where $t_{dwell}$ is the Dwell Time ($t_{dwell}$), $N_{vertices}$ the number of vertices on the TCP, and $N_{TCP}$ the number of TCP *Pattern Repetitions* (*patRepeat*). Additionally, a photonic signal can be integrated at a central spot to calculate the Center-Frequency-Ratio. After completing the TCP, including the optional center spot, the beam is placed back in the initial position. Given its quantized and repetitive nature, we call the entire completed procession of signal integration a TCP *cycle* or round-trip.

After completing a TCP, the Effective Frequency at Offset (EFO) is calculated by dividing the number of detected photons (ECO) by the $t_{dwell}$. It is important to point out that the current version of the microscope does not report the raw photon counts recorded on the vertices of the TCP, which sensitively limits the scope of post-processing. Both ECO and EFO refer to the overall photons collected during position estimation.

If the *Automatic Background Estimation* (*bgcSense*) has been enabled, the local estimated background signal is subtracted from the EFO. The EFO is then compared against the *Background Threshold* (*bgcThreshold*) to determine if the integrated signal can be considered valid.

If the EFO is below the Background Threshold, another round trip is kicked off, and the collected photons are discarded.

If the EFO surpasses the *Background Threshold*, the collected photons are counted against the Photon Limit (*PL*), i.e., the minimal number of photons requested to be used for a localization estimation.

Should the number of detected photons above the background threshold collected be less than the *PL*, another *TCP cycle* is started, and the photons retained.

Should the number of valid photons collected be equal or surpass the *PL*, the MINFLUX microscope will start to calculate a next localization estimate, which after being corrected in a second step using the *Estimation Coefficients* (*estCoefficients*) is used as the initial particle position for the next run. A Galvanometric Scanner is used to re-center the virtual circle around that position, and the routine is re-engaged (Fig. 1d).

Should the number of valid photons remain insufficient for a number of *cycles* determined by the *Single Interval Linger Time* (*maxOffTime*) and by the number *of Permitted Localization Attempts* (*stickiness*), the particle is considered lost and the SPT routine restarted (Fig. 1b, c, d).

The employed parameter controlled *MFX*-enabled SPT routine as provided by the manufacturer (*Abberior Instruments* GmbH.) is reported in ref. 9 and used in refs. 28,29. A detailed overview of the sequence parameters is found in Supplementary Note 1 and the GitHub repository provided (see Data availability).

Our experimental hardware highlights that moving the *doughnut-beam* center between vertices takes about $t_{EOD} \approx 5.3\mu s$, while the localization estimation and TCP repositioning take about $t_{REP} \approx 20\mu s$. This enables us to write down an ideal time-to-localization $t_{loc}^{2D}$ for a single *MFX* determined particle localization in 2D:

$$t_{loc}^{2D}(\eta) = \eta \cdot (t_{\text{dwell}} \cdot (1 + \delta_{\text{CFR}} \cdot \gamma_{\text{CFR}}) + N_{\text{TCP}} \cdot t_{\text{EOD}} \cdot (N_{\text{vertices}} + \delta_{\text{CFR}})) + t_{\text{REP}} \tag{4}$$

$$t_{loc}^{2D}(\eta) = \eta \cdot \chi(t_{\text{dwell}}) + t_{\text{REP}} \tag{5}$$

Where $\eta \in [1, \infty)$ is the number of roundtrips completed, $\delta_{\text{CFR}}$ is a delta-function that returns $\delta_{\text{CFR}} = 1$ if the CFR-Check is enabled, and $\gamma_{\text{CFR}}$ is a factor that determines the amount of time spent integrating the signal for the *CFR-Check* (*ctrDwellFactor*). We summarize the constant part of Eq. 4 into $\chi(t_{\text{dwell}})$ in Eq. 5.

To display the split between signal integration $t^{\eta}_{\text{signal}}$ and hardware overhead $t^{\eta}_{hw}$, we can express Eq. 4 in the following way:

$$t^{2D}_{loc}(\eta) = \eta \cdot t_{dwell} + (\eta \cdot (t_{dwell} \cdot \delta_{\text{CFR}} \cdot \gamma_{\text{CFR}} + N_{\text{TCP}} \cdot t_{\text{EOD}} \cdot (N_{\text{vertices}} + \delta_{\text{CFR}})) + t_{\text{REP}}) \tag{6}$$

$$t^{2D}_{loc}(\eta) = t^{\eta}_{\text{signal}} + t^{\eta}_{hw} \tag{7}$$

### Temporal optimization

Considering linear Eq. 5, we determine two angles of minimization, the number of TCP round trips $\eta$ and the slope $\chi(t_{\text{dwell}})$. While minimizing the latter is trivial considering Eq. 4, we must understand $\eta$ as a function of the laser power, *PL*, and $t_{dwell}$ that minimizes exactly when at least *PL* number of photons arrive at the detector within $t_{dwell}$ seconds. Providing a complete mathematical model of this behavior is beyond the scope of this work, but for our purposes, it is sufficient to understand that these three factors, i.e., laser power, *PL*, and $t_{dwell}$, play an essential role when trying to optimize *MFX*-enabled SPT experiments.

### Turning angle distribution

Turning angles are defined as the angle between the displacement vectors $\vec{a}$ and $\vec{b}$, which describe the particle motion along three consecutive positions A, B, C. It is calculated as follows:

$$\cos(\phi) = \frac{\vec{a} \circ \vec{b}}{|\vec{a}| \cdot |\vec{b}|} \tag{8}$$

The turning angle distribution (TAD) can be used to investigate directionality between successive datapoints, thereby posing as a tool to describe single-particle motion across time.

### Extracting the number of TCP cycles per localization

Assuming a steady fluorescent signal, we can straightforwardly extract the number of *cycles* $\eta$ required for each localization in *MFX* from the data provided as follows:

$$\eta = \frac{N}{\psi_{\text{MFX}} \cdot t_{\text{dwell}}} = \frac{N}{\frac{N}{\eta \cdot t_{dwell}} \cdot t_{\text{dwell}}} = \frac{\text{ECO}}{\text{EFO} \cdot t_{\text{dwell}}} = \eta \tag{9}$$

Where $N$ (ECO) is the number of photons acquired at the TCP during position estimation, $\psi_{\text{MFX}}$ (EFO) is the photon detection frequency at the TCP and $t_{\text{dwell}}$ is the dwell time as put in the sequence.

### Ergodic hypothesis

The ergodic hypothesis, the only fundamental assumption in equilibrium statistical physics[31], implies that the average of a process parameter over time would equate the average over the complete statistical ensemble. In the case of particle diffusion, this relates to:

$$\varepsilon = \frac{D_{e\text{MSD}}}{D_{t\text{MSD}}} \tag{10}$$

Where $D_{t\text{MSD}}$ is the time-average diffusion coefficient and $D_{e\text{MSD}}$ is the ensemble-average diffusion coefficient.

### Defining an upper limit for trackable diffusion rates

Assuming Brownian particles to exhibit a constant average photon emission rate $\langle \theta_{\text{P}} \rangle$ and the average photon detection rate $\langle \psi_{\text{MFX}} \rangle \leq \langle \theta_{\text{P}} \rangle$, we can

give an estimation of the upper limit of possibly trackable diffusion rates in 2D $D^{2D}_{\text{max}}$ given $\sigma^{d=2}_{\text{Diffusion}}$ and that the particle needs to remain within the area of the TCP during the localization process.

$$(L/2)^2 \pi = \pi (\sigma^{d=2}_{\text{Diffusion}})^2 \tag{11}$$

$$L^2 = 16 \cdot t^{loc}_{\min} D^{2D}_{\max} | \min(t^{2D}_{loc}(\eta)) = t^{2D}_{loc}(\eta = 1) \tag{12}$$

$$D^{2D}_{\max} = \frac{L^2}{16} \cdot \frac{1}{t^{\eta=1}_{\text{idealsignal}} + t^{\eta=1}_{\text{hw}}} | t^{\eta=1}_{\text{idealsignal}} = \frac{N_{\text{PL}}}{\langle \theta_{\text{P}} \rangle} \tag{13}$$

$$D^{2D}_{\max} = \frac{L^2}{16} \cdot \frac{\langle \theta_{\text{P}} \rangle}{N_{\text{PL}} + \langle \theta_{\text{P}} \rangle t^{\eta=1}_{\text{hw}}} \tag{14}$$

$$D^{2D}_{\max} \leq \frac{L^2}{16} \cdot \frac{\langle \psi_{\text{MFX}} \rangle}{N_{\text{PL}} + \langle \psi_{\text{MFX}} \rangle t^{\eta=1}_{\text{hw}}} \tag{15}$$

$$D^{2D}_{\max} \leq \frac{L^2}{16} \cdot \frac{\langle \psi_{\text{MFX}} \rangle}{N_{\text{PL}} + \langle \psi_{\text{MFX}} \rangle \cdot 52 \mu s} \quad | \text{ for hexagonal TCP} \tag{16}$$

$$D^{2D}_{\max} \leq \frac{L^2}{16} \cdot \frac{\langle \psi_{\text{MFX}} \rangle}{N_{\text{PL}} + \langle \psi_{\text{MFX}} \rangle \cdot 36 \mu s} \quad | \text{ for triangular TCP} \tag{17}$$

### Statistics and reproducibility

All values in Table 2 and Supplementary Table 1 are provided as described in the respective captions. Ensemble-average MSD curves were calculated across the entire dataset as the mean MSD value per time lag. The turning angle distributions were calculated as the bin-average of all turning angles between [0°,180°] with a bin size of 5°. All boxplots displayed were generated using the *matplotlib* (v3.9.2)[32] Python library. Their configurations are found in the respective captions. Tracking and linkage of TIRF SPT data was performed using the *trackpy* (v0.6.4)[33] Python library. The sample sizes *n* found in Table 2 refer to the number of individual tracks per dataset. All MINFLUX datasets corresponding to a single parameter investigation, i.e., where only one parameter is changed between experiments, are taken on the same sample. If not stated otherwise in the caption, all single-parameter investigations were performed on biologically independent samples. The TIRF SPT reference data was taken on a single sample.

### Reporting summary

Further information on research design is available in the Nature Portfolio Reporting Summary linked to this article.

## Code availability

Due to its high relevance to ongoing projects, we decided to postpone releasing our custom code on GitHub to a later date. Until then, parts of the code can be made available upon specific request. Please, contact B.T.L.V. (bela.vogler@uni-jena.de).

## Data availability

All data and sequences used in the manuscript are made available within the Parameter-Optimization-for-MINFLUX-Microscopy repository on GitHub. The provided data includes the raw information extracted from the proprietary file structure (Abberior Instruments GmbH) returned by the MINFLUX in a streamlined format, the raw TIRF SPT trajectories, the processed MINFLUX and TIRF SPT trajectories including MSD and metadata, as well as the numerical source data for all graphs and charts found in this work. Files are named based on their respective experimental parameters. We include a toolbox to conveniently access and format the data. The raw MINFLUX files in NPY-format, as exported from *MINFLUX-IMSPECTOR* (commercial version-16.3.15645-m2205), as well as the raw TIRF image stacks, are made available within a Zenodo repository (*Data*

*Deposition-Parameter Optimization for MINFLUX Microscopy enabled Single Particle Tracking*; https://doi.org/10.5281/zenodo.17153525[34]). Access to the relevant *MINFLUX-IMSPECTOR* MSR files can be made available upon specific request. Please, contact B.T.L.V. (bela.vogler@uni-jena.de).

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

## Acknowledgements

The authors greatly acknowledge financial support by the Deutsche Forschungsgemeinschaft (DFG, German Research Foundation; Instrument funding MINFLUX Jena INST 275_405_1; Germany´s Excellence Strategy —EXC 2051—Project-ID 390713860; project number 316213987—SFB 1278; GRK M-M-M: GRK 2723/1—2023—ID 44711651), the State of Thuringia (TMWWDG), the Leibniz Association (Leibniz ScienceCampus InfectoOptics Jena financed by the funding line Strategic Networking of the Leibniz Association, project number W8/2018; and Leibniz Collaborative Excellence Program, project AMPel—project numer K548/2023), the Free State of Thuringia (TAB; Advanced Flu-Spec/2020 FGZ: FGI 0031), and the Alexander von Humboldt Foundation (Research Group Linkage Fund). Further, this work is supported by the BMFTR (Federal Ministry of Research, Technology, and Space), funding program LIVE2QMIC (FGZ: 13N15956), as well as Photonics Research Germany (FKZ: 13N15713 / 13N15717) and is integrated into the Leibniz Center for Photonics in Infection Research (LPI). The LPI initiated by Leibniz-IPHT, Leibniz-HKI, UKJ, and FSU Jena is part of the BMFTR national roadmap for research infrastructures.

## Author contributions

Conceptualization: B.T.L.V., F.R.; Methodology: B.T.L.V., F.R., and Z.Z.; Investigation: B.T.L.V., F.R., G.D.A., and Z.Z.; Visualization: B.T.L.V., F.R.; Supervision: F.R., C.E.; Writing—original draft: B.T.L.V., F.R., G.D.A., and Z.Z.; Writing—editing and review: B.T.L.V., F.R., and C.E.

## Funding

## Competing interests

The authors declare no competing interests.
