## [Transparent Peer Review file · Communications Biology]

Parameter Optimization for MINFLUX Microscopy enabled Single Particle Tracking

Corresponding Author: Professor Christian Eggeling

Version 0:

Reviewer comments:

Reviewer #1

(Remarks to the Author)

Vogler et al. performed an analysis of the performance of a commercial MINFLUX microscope for single particle tracking measurements. Their results are indeed interesting as they show that if the acquisition parameters are not properly set, it is easy to obtain artifacts, as exemplified in the artificial preferential directions detected in the Turning Angle Distribution (TDA) shown in Figure 2.

MINFLUX is a powerful method that relies as much on the measurement as on the analysis of the data. Taking into account the impact of other works using MINFLUX for single molecule tracking, I think the findings exposed in this work are highly relevant. However, the manuscript in its present form appears to me as premature, mainly due to the following three reasons:

1. The paper alleges that it provides a parameter optimization strategy. However, this is not clearly stated. Instead, the influence of various parameters is evaluated. A clear step-by-step optimization should be provided, ideally illustrated with examples
2. The parameters involved are not clearly explained (e.g. stickiness, MaxOffTime, Galvo-damping, etc.). I understand this measurements are performed with a commercial instrument, which its functioning is not open. This “black box” instruments could a danger to obtaining reliable data y evaluations like the one provided in this work are highly relevant. Nonetheless, a better understanding and description of the parameters involved would be beneficial.
3. It would be interesting to perform the tests on a well characterized system, so as to have a ground truth to compare to. I understand that MINFLUX aims at exploring ranges of spatio-temporal resolution that are hard/impossible to reach with other methods, but measurements could be performed on ranges of spatio-temporal resolution compatible with other methods (e.g. camera based methods) to test for artifacts.

Minor issues

4. I could not find Equation (1) in the cited reference [7]. More importantly, this equation is for 1D and it does not offer a clear view of the influence of the parameters. It may be more illustrative to accompany the discussion with some plots or simulations.
5. “MINFLUX microscopy localizes single isolated fluorescent emitters by orbiting an excitation beam with a central intensity minimum (i.e., a doughnut) along a pre-defined path”. In fact, MINFLUX in its original form does not “orbit” the beam. It rather performs a minimal scan. This point has been clarified in a paper of 2022, where different configuration were evaluated doi.org/10.1016/j.bpr.2021.100036
6. “While a slight excess of collected photons leads to a more precise localization, a significant underestimation of the photon limit causes errors in the unbiasing step.” What errors exactly? Please explain. What is an “excess of collected photons”? Any fluorescence method for single-molecule/particle localization should benefit from more collected photons.
7. “. Since the estimator does not consider the time at which each photon is detected [10], all fragments are treated as if they happened in parallel though they are consecutive”. This statement is unclear. How would it help knowing the arrival time of

each photon?

8. "Fewer cycles per localization mean a higher frequency of position updates, which permits following faster particles at a reduced distance from the TCP center leading to more precise tracking." This statement needs further clarification. Fewer cycles per localization, for any given emission rate, means less precise localization due to the fewer detected photons. So, the tracking will be less precise. However, if the goal is determining the diffusion coefficient of a Brownian particle, then, having a larger number of less accurate positions is more efficient.

9. It is not necessary to include an "average emission rate", an "average photon detection frequency" and a "lossless detection" in the treatment. All that matters is the average photon detection rate. This could be improved with a better emitter and/or a better setup. Assuming a lossless detection, or any detection efficiency explicitly is irrelevant. It is equivalent to assuming a given quantum efficiency of the emitters.

Reviewer #2

(Remarks to the Author)

Vogler et al. present a discussion around MINFLUX based single particle tracking (SPT) and the optimization of acquisition parameters. They deduce a theoretical upper limit for diffusion that can be reliably tracked and investigate the effects varying tracking parameters experimentally and theoretically. MINFLUX-based tracking is different from traditional SPT approaches as the data are processed on the fly and the microscope readily provides localizations. In traditional SPT, localizations are obtained mostly from post-processing. Therefore, optimization of tracking acquisition parameters is paramount to a successful tracking experiment using MINFLUX.

The manuscript is aimed at a technical audience who is using MINFLUX for tracking of free diffusion. Nevertheless, the presented results and considerations about parameter optimization are solid and important to that community. I am reviewing this manuscript from the point of view of someone who has experience in measuring diffusion dynamics and has a large interest in MINFLUX but is not using it regularly (yet). With the rise in popularity of MINFLUX in general, the study could have important implications. However, I would suggest increasing clarity in writing and data presentation as outlined below to make the manuscript more accessible to a broader audience.

In summary before I can support publication, I suggest a few textual clarifications/corrections, increasing the number of experiments to at least three independent bilayers (Figure 2c,d), and adding more discussion (consider adding a discussion section). See detailed comments below.

1) Line 26/27 same in line 30 referring to Figure 1a about the parameter L. L is not illustrated in Figure 1a.

2) Suggestions for Figure 1a: Add more descriptions and make it consistent with Table 1

- Add 1st iteration, 2nd iteration etc titles
- Add L and v on the schematic
- Table 1 outlines L as 302 nm for the third iteration. Why is the third iteration smaller than first and second (with 284 nm) on the Figure?
- If L is radius of TCP and L of fourth iteration is 150 nm, the scale bar bottom right of Figure 1a is about two times too big.
- It might be helpful to indicate on the schematic that only the fourth iteration is investigated in this study. For example, divide into "Find particle" and "Track particle"
- Consider adding the timings (dwell times, tloc, overhead) on the schematic.

3) Line 46: "cease of operations" is an odd choice of words, consider rephrasing.

4) It is not clear how PL and dwell time dT can be independently varied.

Line 51 indicates that PL is set and Line 52-53 indicate that dT is predefined. So what happens if PL is reached (or not reached) within dT. Is the localization discarded if number of photons < PL within dT? If PL is reached with tloc < dT what happens during the remaining time? This should be explained better as this is vital for the parameter optimization.

5) Line 68 refers to Figure 1b and discusses the influence of particle motion on the localization precision. It would be helpful to write out how that looks for an immobile emitter. In other words that in that case distance to TCP center is always 0. When are deviations apparent? From the plot in Figure 1b it seems at around 20 nm from center for L75 and around 50 nm for L150.

6) Figure 1c: The caption speaks about the fraction of TCP area covered. The graph shows TCP area coverage in %. The background colors indicate L75, L150, and >L150. That is a little confusing as for TCP coverage in fraction or percentage the TCP area should be fixed based on a certain L. The message of the plot is clear and important as the motion of the particle at a given L and D limits tracking abilities.

7) Line 343 states "solid line L=100nm", panel 1b and text indicate that this should be 150 nm.

8) Presentation of experimental results in Figure 2c,d needs clarifications.

- What is the ground truth of diffusion coefficient? This could be obtained from a different method ie FRAP TIRF-SPT or FCS?
- For readability it would help to include PL and dT on the plots as titles

- 3c right panel (PL 50 and dT 100) and 3d middle panel (PL 50 and dT 100): Why are the standard deviations and mean values so different?

- The caption indicates that all PL and dT comparisons have been performed on the same sample, N=1 each. It would be helpful to compare measurements from three or more SLBs to understand if the observed effects are a) reproducible and b) matter i.e. are larger than sample to sample variability.

- The caption for c and d indicate "various" PL and dT. Consider saying "three"

9) The authors refer to an optimization strategy. They show theoretically the effect of L, D, and dT and experimentally of dT and PL. This feeds into eq 2/3 for the estimation of an upper limit for diffusion coefficients that can be tracked. As the title of the paper states "Parameter Optimization...", the authors should consider providing the reader and other MINFLUX users with a set of rules how exactly to go about the optimization. Providing a clear strategy would increase the impact of the work.

10) Figure 3b nicely relates experimental findings (Figure 2c) with the theoretical predictions. The manuscript would benefit from further discussion for other experimental conditions. For example, in the case of PL=20 and dT=100 (Figure 2c): Theory in Figure 3a suggests that at a photon emission frequency of 100 kHz upper trackable limit is about 5 $\mu\text{m}^2/\text{s}$. The experiment finds 0.74 $\mu\text{m}^2/\text{s}$ (Figure 2c). So, can this number be trusted? This also reiterates the point about needing a ground truth for the diffusion coefficient in the SLB.

11) Line 215: Missing word: "are reported in Table 1"

12) Line 233,244: Why are these equations not numbered?

13) Schmidt et al (PMID: 33674570) perform tracking on a similar setup and sample (DPPE-Atto647N in DOPC:Chol (50/50)). They change the pattern from hexagonal to triangle for the final step in tracking to increase temporal resolution. Can you discuss and relate your results? What if you changed to triangle pattern with a smaller area than hexagonal/circular?

14) MINFLUX tracking had exceptional success for investigating processive motion such as a motor walking on a filament (PMID: 36893247, PMID: 38811803). For the presented results the authors assume free diffusion within the TCP for final localization. How would the results be affected by processive motion within the TCP?

Version 1:

Reviewer comments:

Reviewer #1

(Remarks to the Author)

The revised version of the manuscript by Vogler et al addresses most of the comments made in the first round of review. Particularly, the authors now mention this work is about optimizing the function of a particular commercial instrument, and provide a more detailed explanation of the parameters used.

I still have the following concerns about this work:

1) Overall, the language and the organization of the information is cumbersome. The paper is hard to understand.

2) The text should be revised thoroughly to remove unnecessary adjectives, vague statements and non-logical cause-consequence statements. Just to give an example from the first paragraph:

"For creating a single particle trajectory, iMFX necessarily performs particle position estimation and trajectory linkage at runtime. This is in stark contrast to conventional SPT applications, where analysis of this kind usually happens in post processing. Thus, whatever data is returned by the microscope is an interpretation of the signal produced by the sample based on the given acquisition parameters like L, N or v. This allows to seamlessly follow single emitters in time over all three dimensions of space with high spatiotemporal resolution"

- "necessarily" is unnecessary

- "whatever data"? The data to be retrieved are positions over time.

- "This allows to seamlessly follow single emitters in time over all three dimensions of space with high spatiotemporal resolution". A possibility of seamlessly following a single emitter does not logically emerge from the previous description. Are all the parameters mentioned really necessary to understand the measurement pipeline? It seems to me some are not really necessary and some are repeated, like dT and t_{dwell} (I could not understand their difference).

What are the exclamation points in equation (2)?

3) It is not clear to me the role of the iterative reduction of L. The paper title and first paragraphs mention the iterative MINFLUX approach, which is based on iteratively reducing L to achieve a higher localization precision. However, later the authors state:

"Here, we focus on engineering only the last iteration of the full sequence needed to obtain a valid localization as it is the most relevant for tracking applications"

And, if I understood correctly, in all experiments L was a predefined fixed parameter. So all the optimization performed does not involve any iterations of the size of L. Is this correct?

Also in the section "Iterative MINFLUX Localization Estimation" there is no mention about an iterative reduction of L.

So, is this work about SPT with an iterative MINFLUX approach or not?

If not, the title and text should be clear about it.

4) It is not clear whether the authors correct the trajectories in post-processing by re-analyzing the photon counts with the numerical estimators based on the actual shape of the excitation beam, as it was done in the original MINFLUX publication. If not, why not?

If yes, how was the shape of the excitation beam determined?

5) Related to the previous comment, I suggest the authors to make a comparison of their tracking pipeline to the previously reported tracking experiments with MINFLUX.

6) "Summarizing our theoretical and experimental considerations, it is evident that minimizing the time between consecutive localizations, while keeping PL as high as possible, is key to enabling high-fidelity iMFX-enabled SPT measurements". Isn't it an obvious statement for any SPT method?

7) "The Turning Angle Distribution (TAD) demonstrates an additional geometric bias in that a set of directions is preferred with decreasing PL (Figure 2b, $dT = 100\mu s$) digitizing the direction of positional updates, i.e. too low values of PL resulted in a biased detection of diffusion. This is a direct result of the reduced number of acquired photons and the thus reduced size of the statistical data sample available for analysis."

I think the directional bias is primarily a result of the geometry of the TCP. Then a low photon count could be detrimental. If not, please explain.

Related to this point, the authors state:

"Additionally, it may seem a plausible approach to increase the temporal resolution by adopting TCP scanning patterns with fewer positions. This has already been employed in the past [7, 8], and it does indeed marginally increase the sampling rate, since fewer movements of the laser must be performed during one TCP, and thus less time is spent on one acquisition cycle. As we have shown (Figure 2b), this again results in an angular bias in the position estimator, which may be undesirable. Therefore, adopting a hexagonal TCP is preferable, at least when probing diffusive motion with a high degree of randomness[8]"

But the hexagonal TCP also shows directional bias, or not? Was not the data shown in figure 2b obtained with the hexagonal TCP?

Reviewer #2

(Remarks to the Author)

With the revised version of their manuscript and the point-by-point responses, the authors have addressed all my comments. They have extensively improved the manuscript, added explanations and data from TIRF-SPT as comparison to iMFX-SPT.

I appreciate the addition of the guide and descriptions in Appendix I and II making these parameters and their optimization accessible for all MINFLUX users. Consider mentioning the appendices at the beginning of the manuscript to encourage the reader to take a look early on.

Minor comments to consider:

Line 70: Full stop missing

Line 112: Shouldn't the area of the TCP circle be $\pi(L/2)^2$?

Line 133 and Figure 2a,b: What is the concentration of DSPE-PEG-Biotin in the SLB? The text states "SLB, 100% DOPC".

The revised manuscript reads well, and I am fully supportive of publication. Congratulations on an important piece of work for the tracking and MINFLUX community.

Version 2:

Reviewer comments:

Reviewer #1

(Remarks to the Author)

The authors have addressed most issues raised by the reviewers, and I can recommend its publication. Nonetheless, the paper is still not fully clear about the role of iterations to reduce L. In their rebuttal letter, the authors have explained why they focus on the later stage of the localization procedure, which is not iterating to reduce L. However, the manuscript is still misleading in parts, including (unnecessarily) the word "iterative" not only in the title but in all sub-titles and throughout the text. That said, I no longer wish to discuss this issue and leave it to the authors to consider clarifying this point in the final version of the paper.

Responses to Reviewers

We thank all the reviewers for their time and commitment to read our manuscript in detail and to make important comments. We have considered them all and have revised the manuscript accordingly. Specifically, we added a lot more details to text and supplements. This helped to improve the quality of our manuscript significantly. Below are the detailed responses to the reviewers' comments.

Reviewer I:

1 The paper alleges that it provides a parameter optimization strategy. However, this is not clearly stated. Instead, the influence of various parameters is evaluated. A clear step-by-step optimization should be provided, ideally illustrated with examples.

- We agree and have hence added a comprehensive *Guide for Iterative-MINFLUX Parameter Optimization for 2D Single Particle Tracking Measurements* as Appendix I.

2 The parameters involved are not clearly explained (e.g. stickiness, MaxOffTime, Galvo-damping, etc.). I understand this measurements are performed with a commercial instrument, which its functioning is not open. This “black box” instruments could a danger to obtaining reliable data y evaluations like the one provided in this work are highly relevant. Nonetheless, a better understanding and description of the parameters involved would be beneficial.

- We agree and have hence added a comprehensive *Overview of the Most Important Parameters Involved in Iterative-MINFLUX* as Appendix II.

3 It would be interesting to perform the tests on a well characterized system, so as to have a ground truth to compare to. I understand that MINFLUX aims at exploring ranges of spatio-temporal resolution that are hard/impossible to reach with other methods, but measurements could be performed on ranges of spatio-temporal resolution compatible with other methods (e.g. camera based methods) to test for artifacts.

- We agree with the reviewer on the benefit of using a concrete reference value taken using a camera-based system to better gauge the influence of certain parameters on the MINFLUX measurement. We have hence repeated our MINFLUX SPT experiments as well as taken SPT data using a TIRF-based microscope on Supported Lipid Bilayers (SLB) patches generated out of a Giant-Unilamellar-Vesicle (GUV). On the one hand we could confirm our results from our previous MINFLUX-based SPT measurements on SLBs generated through microfluidics. On the other hand, the TIRF-based SPT experiments were important to strengthen our discussion on the right choice of experimental MINFLUX parameters (see Figure 2).

4 I could not find Equation (1) in the cited reference [7]. More importantly, this equation is for 1D and it does not offer a clear view of the influence of the parameters. It may be more illustrative to accompany the discussion with some plots or simulations.

- We added a more precise reference to locate the relevant equation, given that it is in the (quite extensive) supplementary material of that reference.
- We highlight the influence of the PL and TCP diameter L in Figure 1e (previously Figure 1b).

- While of course this equation is reductive, we found it useful to include it as a reference to understand the variables at play in the determination of the localization precision, namely the TCP diameter L , number of photons N , and most importantly the distance between the center of the TCP and the fluorophore x . The localization precision of the SPT experiment is ultimately derived from the data, as we now elaborate more precisely in the manuscript. Further discussions on the CRLB would lengthen the manuscript significantly and divert from our main topic and is therefore beyond the scope of this work.

5 “MINFLUX microscopy localizes single isolated fluorescent emitters by orbiting an excitation beam with a central intensity minimum (i.e., a doughnut) along a pre-defined path”. In fact, MINFLUX in its original form does not “orbit” the beam. It rather performs a minimal scan. This point has been clarified in a paper of 2022, where different configuration were evaluated doi.org/10.1016/j.bpr.2021.100036

- We have resolved this point by specifying in the text what exactly the MINFLUX does when it performs its pattern scan, which indeed is different from an orbital scan in related techniques. We hope this clarifies the mechanism of MINFLUX scanning satisfactorily.

6 “While a slight excess of collected photons leads to a more precise localization, a significant underestimation of the photon limit causes errors in the unbiasing step.” What errors exactly? Please explain. What is an “excess of collected photons”? Any fluorescence method for single-molecule/particle localization should benefit from more collected photons.

- Of course, we agree with the reviewer that, in principle, more photons will lead to a better localization precision. In the case of *IMFX*, this is not quite as simple as in other techniques, since the number of photons that contribute to the final localization precision is fixed as a parameter (Photon Limit, PL). Additional photons above this value offer diminishing returns, since the estimator algorithm is optimized for the PL , and cannot be modified, either during or after the acquisition. We now offer a more thorough explanation in the text, hopefully driving better this point.

7 “Since the estimator does not consider the time at which each photon is detected [10], all fragments are treated as if they happened in parallel though they are consecutive”. This statement is unclear. How would it help knowing the arrival time of each photon?

- We apologize for the misunderstanding caused by this sentence. We aimed here to further the point that it is not possible to know the exact behavior of a diffusing particle during a TCP iteration, since the estimator produces a localization based only on how many photons were collected at which point of the TCP and makes a guess of the most likely position of the particle within the TCP itself. It was our intention to hint at the fact that inserting in the estimator the information of the point in time, as well as the TCP location, at which the device collects photons, could better inform on how the particle is moving and follow it more efficiently. This would be similar to how orbital tracking works, although we recognize that in this case it would be more challenging, and not necessarily desirable. We expanded on this point in the text to add more clarity.

8 “Fewer cycles per localization mean a higher frequency of position updates, which permits following faster particles at a reduced distance from the TCP center leading to more precise tracking.” This statement needs further clarification. Fewer cycles per localization, for any given emission rate, means less precise localization due to the fewer detected photons. So, the tracking

will be less precise. However, if the goal is determining the diffusion coefficient of a Brownian particle, then, having a larger number of less accurate positions is more efficient.

- The optimization strategy highlighted by the reviewer is valid, and in fact is the strategy adopted in the majority of SPT techniques. In MINFLUX, however, the issue is that one does not perform the localization on a whole field of view, but rather on an extremely small scale. Therefore, spending more than one cycle per localization may in fact worsen the localization precision despite the larger number of photons collected, due to the particle exploring a larger portion of the TCP. This will happen in a way that is difficult to ascertain, since the device will still output a localization (provided the particle does not escape the TCP in the meantime). We added detail to the text to clarify this point.

9 It is no necessary to include an “average emission rate”, an “average photon detection frequency” and a “lossless detection” in the treatment. All that matters is the average photon detection rate. This could be improved with a better emitter and/or a better setup. Assuming a lossless detection, or any detection efficiency explicitly is irrelevant. It is equivalent to assuming a given quantum efficiency of the emitters.

- We agree with the reviewer that distinguishing between these quantities may lead to confusion in the reader, so we now only discuss of average photon detection rate for simplicity.

Reviewer II:

1 Line 26/27 same in line 30 referring to Figure 1a about the parameter L. L is not illustrated in Figure 1a.

- We thank the reviewer and have hence added a reference to the TCP diameter L in Figure 1b. Additionally, we have corrected two instances where L was referred to incorrectly as the radius of the TCP instead of the diameter.

2 Suggestions for Figure 1a:

- Add more descriptions and make it consistent with Table 1
 - Add 1st Iteration, 2nd iteration etc titles
 - Add L and v on the schematic
 - Table 1 outlines L as 302 nm for the third iteration. Why is the third iteration smaller than first and second (with 284 nm) on the Figure?
 - If L is radius of TCP and L of fourth iteration is 150 nm, the scale bar bottom right of Figure 1a is about two times too big.
 - It might be helpful to indicate on the schematic that only the fourth iteration is investigated in this study. For example, divide into “Find particle” and “Track particle”
 - Consider adding the timings (dwell times, tloc, overhead) on the schematic.
- The reviewer has brought up some important refinements for Figure 1 and the corresponding Table 1. We tried to integrate as many suggestions as possible while maintaining a clean and easily interpretable structure throughout the panels. With this, we hope to have achieved a more comprehensive overview of the Iterative MINFLUX Single Particle Tracking process.

3 Line 46: “cease of operations” is an odd choice of words, consider rephrasing.

- We followed the recommendation of the reviewer and made a different choice of words.

4 It is not clear how PL and dwell time dT can be independently varied. Line 51 indicates that PL is set and Line 52-53 indicate that dT is predefined. So what happens if PL is reached (or not reached) within dT. Is the localization discarded if number of photons < PL within dT? If PL is reached with tloc < dT what happens during the remaining time? This should be explained better as this is vital for the parameter optimization.

- We have added two paragraphs in the Methods-section that elaborate the Iterative-MINFLUX Localization Estimation Process in greater detail. These points are now further explained in the revised supplementary material.

5 Line 68 refers to Figure 1b and discusses the influence of particle motion on the localization precision. It would be helpful to write out how that looks for an immobile emitter. In other words that in that case distance to TCP center is always 0. When are deviations apparent? From the plot in Figure 1b it seems at around 20 nm from center for L75 and around 50 nm for L150.

- We added some lines at the end of the same section to clarify this issue. Essentially, this is quite difficult to express for every case, as the emergence of significant deviations from the CRLB in the case of SPT experiments is dependent on several concurrent factors, most notably the diffusion rate, but e.g. also varying background. This can however be compensated, as the reviewer notices, by larger values of L, which have a shallower gradient for the CRLB compared to smaller values, as is evident from Figure 1e (previously Figure 1b). We have highlighted this in the text.

6 Figure 1c: The caption speaks about the fraction of TCP area covered. The graph shows TCP area coverage in %. The background colors indicate L75, L150, and >L150. That is a little confusing as for TCP coverage in fraction or percentage the TCP area should be fixed based on a certain L. The message of the plot is clear and important as the motion of the particle at a given L and D limits tracking abilities.

- We agree with the reviewer that including multiple TCP diameters in the background color convolutes the message of the graph. Hence, we simplified the respective panel (now Figure 1f) in hopes of clarification.

7 Line 343 states “solid line L=100nm”, panel 1b and text indicate that this should be 150 nm.

- This has now been corrected. We thank the reviewer for pointing this out.

8 Presentation of experimental results in Figure 2c,d needs clarifications.

- We thank the reviewer for the comprehensive suggestions to improve the clarity of what has originally been Figure 2c, d. We took inspiration from the comment and modified what is now Figure 2c.

- What is the ground truth of diffusion coefficient? This could be obtained from a different method ie FRAP TIRF-SPT or FCS?

- We added a TIRF SPT measurement taken on a similar sample to Figure 2c as a line of reference to Figures 2c-e.

- For readability it would help to include PL and dT on the plots as titles

- The focus-parameters of the respective datasets have been moved to the x-axis of the plot.

- 3c right panel (PL 50 and dT 100) and 3d middle panel (PL 50 and dT 100): Why are the standard deviations and mean values so different?

- This discrepancy between the datasets had been due to them being taken on different samples across different days with different alignments. We have repeated the experiments and marked sets taken using the same sequences accordingly.

- The caption indicates that all PL and dT comparisons have been performed on the same sample, N=1 each. It would be helpful to compare measurements from three or more SLBs to understand if the observed effects are a) reproducible and b) matter i.e. are larger than sample to sample variability.

- We agree that a comparison between SLBs and datasets would be beneficial. Therefore, Figure 2c displays the results of three samples taken with PL50 and dt100, two on GUV-Patches and one on a SLB generated by a flow-chamber hosted lipid deposition. We also include an example of the same parameters on the same sample, i.e. DMP0 and PL10. Both cases are marked accordingly.
- The goal of the experiments is to demonstrate the predictable impact parameter changes have on the reported results. Just as we elaborate in the main text, iterative MINFLUX performs all steps of classical SPT *ad hoc*, i.e. during tracking. In this process, each position relies on the existence of a previous one. Selecting sequence parameters that stray too far from the ideal, i.e. that do not allow continuous tracking, will result in the microscope being unable to track most of the particles, i.e. a failed experiment. Therefore, the influence the parameters have ranges from varying reported values as demonstrated in our work, to a failed experiment with no data to report in the first place.

- The caption for c and d indicate “various” PL and dT. Consider saying “three”

- For the sake of clarity, we modified the caption of Figure 2 accordingly.

9 The authors refer to an optimization strategy. They show theoretically the effect of L, D, and dT and experimentally of dT and PL. This feeds into eq 2/3 for the estimation of an upper limit for diffusion coefficients that can be tracked. As the title of the paper states “Parameter Optimization...”, the authors should consider providing the reader and other MINFLUX users with a set of rules how exactly to go about the optimization. Providing a clear strategy would increase the impact of the work.

- We agree and have hence added a comprehensive *Guide for Iterative-MINFLUX Parameter Optimization for 2D Single Particle Tracking Measurements* as Appendix I.

10 Figure 3b nicely relates experimental findings (Figure 2c) with the theoretical predictions. The manuscript would benefit from further discussion for other experimental conditions. For example, in the case of PL=20 and dT=100 (Figure 2c): Theory in Figure 3a suggest that at a photon emission frequency of 100 kHz upper trackable limit is about $5 \mu\text{m}^2/\text{s}$. The experiment finds $0.74 \mu\text{m}^2/\text{s}$ (Figure 2c). So, can this number be trusted? This also reiterates the point about needing a ground truth for the diffusion coefficient in the SLB.

- We briefly discuss the implications of choosing different parameters and hint on possible optimizations beyond what is shown in our work.
- In what is now Figure 3d (former Figure 3b) we compare the result of a dye-tagged SLB SPT experiment, a system that exhibits significantly faster particle diffusion, to the predicted upper limits using the same core parameters as our PL10 benchmark experiments. The reported values ($D_{\text{MSD}}^{\text{max}} \approx 7.5 \mu\text{m}^2/\text{s}$) come significantly closer to the theoretical upper threshold ($D_{\text{max}}^{d=2} \approx 10 \mu\text{m}^2/\text{s}$) but remains below.
- A comparison between the average diffusion rates reported by MINFLUX and TIRF for the QD-tagged SLB samples returns similar results, which leads us to consider our results to be trustworthy. We further compare the results of the dye-tagged SLB to literature with the same conclusion.

11 Line 215: Missing word: “are reported in Table1”

- This has now been corrected. We thank the reviewer for bringing it to our attention.

12 Line 233,244: Why are these equations not numbered?

- This has now been corrected. We thank the reviewer for bringing it to our attention.

13 Schmidt et al (PMID: 33674570) perform tracking on a similar setup and sample (DPPE-Atto647N in DOPC:Chol (50/50). They change the pattern from hexagonal to triangle for the final step in tracking to increase temporal resolution. Can you discuss and relate your results? What if you changed to triangle pattern with a smaller area than hexagonal/circular?

- The comment this reviewer puts forward has a few points to discuss. First: it is indeed true, that a triangular pattern is, for obvious reasons, faster than a hexagonal pattern, given the fewer focal displacements involved there. However, it is shown in the same paper, in Figure 2d and Supplementary Figure S2, and discussed in the text, that using a triangular pattern leads to a localisation estimator bias with a lower degree of angular uniformity. This is obviously undesirable, especially with dealing with diffusive motion with a high degree of randomness on the 2D plane.
- Using a smaller triangular TCP instead of the hexagonal one employed in our work would result in a theoretical increase in localisation precision. However, we maintain that this improvement on paper will be outweighed by the undesirable side effects that follow from our discussion and the previous point, namely that the angular bias in the estimator will be accentuated by the reduced size of the TCP, and that a smaller scanning pattern is easier to be “escaped” by our diffuser.
- To acknowledge the comment, we have added a short discussion of the triangular TCP pattern in the text

14 MINFLUX tracking had exceptional success for investigating processive motion such as a motor walking on a filament (PMID: 36893247, PMID: 38811803). For the presented results the authors assume free diffusion within the TCP for final localization. How would the results be affected by processive motion within the TCP?

- The case brought up by the reviewer is especially relevant, given the most recent applications of MINFLUX. In general, directed or processive motion can be seen as a special, one dimensional, case of our generalized lateral diffusion. In general, the same considerations apply, and the user should adjust the parameters of MINFLUX scanning to ensure that this diffusion motion is well sampled, as we suggest. The reason of this would be, that the process by which the MINFLUX calculates and updates its new position estimation is not affected by the mode of motion, since the system has no pre-conceived notion of the sample. The condition given in Equation 3 would thus work also for particles diffusing in a very specific fashion, given that it is also possible to calculate their diffusion rate.
- We have added this consideration to the text, and we thank the reviewer for pointing this out.

Response to the Reviewers

2nd Review

We thank all the reviewers for their time and commitment reading our manuscript in detail and for raising important comments. As in the previous revision, we tried to consider them all and have revised the manuscript accordingly. Specifically, we were keen on improving the clarity of our arguments and language. The comments raised helped us yet again improve our work decisively.

Below we list the detailed responses to each point raised by the reviewers.

Reviewer I:

The revised version of the manuscript by Vogler et al addresses most of the comments made in the first round of review. Particularly, the authors now mention this work is about optimizing the function of a particular commercial instrument, and provide a more detailed explanation of the parameters used.

I still have the following concerns about this work:

1 Overall, the language and the organization of the information is cumbersome. The paper is hard to understand.

- Together with the important notes raised in the second comment below, we went over our manuscript to improve the clarity of our language and avoid possible causes of confusion. Additionally, we divided the main text into subsections and clearly separated the experimental and theoretical sections to aid the reader navigating the text.

2 The text should be revised thoroughly to remove unnecessary adjectives, vague statements and non-logical cause-consequence statements. Just to give an example from the first paragraph: “For creating a single particle trajectory, iMFX necessarily performs particle position estimation and trajectory linkage at runtime. This is in stark contrast to conventional SPT applications, where analysis of this kind usually happens in post processing. Thus, whatever data is returned by the microscope is an interpretation of the signal produced by the sample based on the given acquisition parameters like L, N or v. This allows to seamlessly follow single emitters in time over all three dimensions of space with high spatiotemporal resolution”

- “necessarily” is unnecessary

- “whatever data”? The data to be retrieved are positions over time.

- “This allows to seamlessly follow single emitters in time over all three dimensions of space with high spatiotemporal resolution”. A possibility of seamlessly following a single emitter does not logically emerge from the previous description.

- We agree with the points raised and have made appropriate changes throughout the text. We thank the reviewer for their appeal to clarity of language.

Are all the parameters mentioned really necessary to understand the measurement pipeline? It seems to me some are not really necessary and some are repeated, like dT and t_{dwell} (I could not understand their difference).

- Indeed, given the complex position estimation routine that iterative MINFLUX employs, we are afraid that all parameters mentioned are necessary to grasp how the microscope operates, in this simplified report.
- dT and t_{dwell} were synonymous. We have now unified them to t_{dwell} throughout all documents to avoid confusion. We thank the reviewer for bringing this to our attention.

What are the exclamation points in equation (2)?

- They were meant to show forced equality. To avoid confusion, we decided to drop the exclamation mark. We thank the reviewer for bringing this to our attention.

3 It is not clear to me the role of the iterative reduction of L . The paper title and first paragraphs mention the iterative MINFLUX approach, which is based on iteratively reducing L to achieve a higher localization precision. However, later the authors state: “Here, we focus on engineering only the last iteration of the full sequence needed to obtain a valid localization as it is the most relevant for tracking applications”

And, if I understood correctly, in all experiments L was a predefined fixed parameter. So all the optimization performed does not involve any iterations of the size of L . Is this correct?

Also in the section “Iterative MINFLUX Localization Estimation” there is no mention about an iterative reduction of L .

So, is this work about SPT with an iterative MINFLUX approach or not? If not, the title and text should be clear about it.

- We understand how some confusion may originate during the reading, especially if the reader is not familiar with MINFLUX in general, or this specific implementation. The device is delivered with a number of pre-defined scanning sequences, all of which involve iteratively shrinking the TCP with decreasing values of L around a localization estimate obtained from the previous step as part of the overall routine.

More specifically, Iterative MINFLUX operates an iterative localization scheme by continuously repeating the same steps. Though the successive reduction of L is a crucial part of the overall SPT routine, given it refines the initial position from which the particle is followed, the underlying localization estimation procedure works the same whether it is a refinement *pattern iteration* (L is being reduced) or a tracking *pattern iteration* (reiterate smallest L).

In our work, we decided to emphasize the optimization of the *final pattern iteration*, i.e. the one with the smallest L , since it is the one from which the final position estimate is obtained, as well as the one that is continuously reiterated during an SPT experiment to track the moving target. Moreover, in our experience the first few *pattern iterations*, (in which L is successively reduced), require less optimization from the values suggested by

the manufacturer, as exemplified by the fact that the diffusion rate we obtain from our MINFLUX experiments match those obtained from more established techniques by only optimizing the final pattern iteration. Nevertheless, our method can also be followed to optimize the rest of the sequence, as the instrument provides the necessary information for this task as well. In our new version of the manuscript, we have been more precise in pointing this out, hopefully clearing the confusion. We thank the reviewer for pointing this out and allowing us to make the text clearer for a non-specialist audience.

4 It is not clear whether the authors correct the trajectories in post-processing by re-analyzing the photon counts with the numerical estimators based on the actual shape of the excitation beam, as it was done in the original MINFLUX publication. If not, why not?

If yes, how was the shape of the excitation beam determined?

- The device we refer to in the present manuscript is different from the original prototype from the paper by Balzarotti et al. In contrast to that, the commercial MINFLUX microscope is designed to directly output the final localizations, which require no post-processing. Even if we wanted, the device does not output the raw data necessary to make post-processing the localizations possible. We added a few sentences to clarify the issue raised by the reviewer in our manuscript (70-78, 440-443) and want to express our gratitude for bringing this topic to our attention.

5 Related to the previous comment, I suggest the authors to make a comparison of their tracking pipeline to the previously reported tracking experiments with MINFLUX.

- We added the following phrases to our manuscript to clarify the issue raised by the reviewer:

The employed parameter controlled iMFX-enabled SPT routine as provided by the manufacturer (Abberior Instruments GmbH.) is reported in [8] and used in [26, 27]. A detailed overview of the sequence parameters is found in Appendix I and the provided GitHub repository (see Data and Sequence availability).

We want to express our gratitude for bringing this topic to our attention.

6 “Summarizing our theoretical and experimental considerations, it is evident that minimizing the time between consecutive localizations, while keeping PL as high as possible, is key to enabling high-fidelity iMFX-enabled SPT measurements”.

Isn't it an obvious statement for any SPT method?

- We agree that it is in principle a straightforward assumption for any SPT method. We found it necessary to be stressed here, since the time between localizations takes an additional meaning in MINFLUX, given that dwell times are in the same order of magnitude as the hardware overhead time. Additionally, as we remark throughout the text, the sampling rate is not set *a priori* by the user but depends on the photon detection rate and the photon limit parameter. In a way, it is not the user, but rather the sample that determines the final detection rate for the SPT experiments, and as such this apparently simple statement implies the optimization steps that we express throughout the text.

7 “The Turning Angle Distribution (TAD) demonstrates an additional geometric bias in that a set of directions is preferred with decreasing PL (Figure 2b, $dT = 100\mu s$) digitizing the direction of positional updates, i.e. too low values of PL resulted in a biased detection of diffusion. This is a direct result of the reduced number of acquired photons and the thus reduced size of the statistical data sample available for analysis.”

I think the directional bias is primarily a result of the geometry of the TCP. Then a low photon count could be detrimental. If not, please explain.

- We agree with the reviewer that the geometry of the TCP has a significant and evident influence on directional bias and have added this remark to the text. Since the specific details of the optimization algorithm are not disclosed by the manufacturer, we cannot point out how exactly how a low photon count affects the bias we observe in the TAD. That being said, we made this claim by relying on our observation that increasing the photon limit reduces directional bias. We thank the reviewer for helping us make this point clearer.

Related to this point, the authors state:

“Additionally, it may seem a plausible approach to increase the temporal resolution by adopting TCP scanning patterns with fewer positions. This has already been employed in the past [7, 8], and it does indeed marginally increase the sampling rate, since fewer movements of the laser must be performed during one TCP, and thus less time is spent on one acquisition cycle. As we have shown (Figure 2b), this again results in an angular bias in the position estimator, which may be undesirable. Therefore, adopting a hexagonal TCP is preferable, at least when probing diffusive motion with a high degree of randomness[8]”

But the hexagonal TCP also shows directional bias, or not? Was not the data shown in figure 2b obtained with the hexagonal TCP?

- We agree with the reviewer and have thus modified the highlighted section from our manuscript to clarify the issue:

“To counteract this effect, it may appear reasonable to adopt TCP scanning patterns with fewer positions to increase the sampling rate. This has already been employed in the past [7, 8], and it does marginally increase the temporal resolution, since fewer positions have to be scanned during one TCP, thus shortening t_{hw}^{η} . However, reducing the number of vertices in the position estimator also results in an undesirable increased angular bias, which is especially true for low numbers of photons (Figure 2b). While in principle, any TCP with a finite number of vertices (i.e. not an ideal circle) will result in some degree of angular bias, the hexagonal geometry has proven to be a suitable compromise between temporal efficiency and angular resolution (compare [8] Figure 2 b-d). Therefore, adopting a hexagonal TCP is preferable, at least when probing diffusive motion with a high degree of randomness [8].”

We want to express our gratitude for bringing this topic to our attention.

Reviewer II:

With the revised version of their manuscript and the point-by-point responses, the authors have addressed all my comments. They have extensively improved the manuscript, added explanations and data from TIRF-SPT as comparison to iMFX-SPT.

I appreciate the addition of the guide and descriptions in Appendix I and II making these parameters and their optimization accessible for all MINFLUX users. Consider mentioning the appendices at the beginning of the manuscript to encourage the reader to take a look early on.

Minor comments to consider:

1 Line 70: Full stop missing:

- This has now been corrected. We thank the reviewer for bringing it to our attention.

2 Line 112: Shouldn't the area of the TCP circle be $\pi(L/2)^2$?:

- This has now been corrected. We thank the reviewer for bringing it to our attention.

3 Line 133 and Figure 2a,b: What is the concentration of DSPE-PEG-Biotin in the SLB? The text states "SLB, 100% DOPC":

- We meant to express that we used almost pure DOPC SLBs. We exchanged the confusing passage with reference to the method section, where we list the Mol% of all lipids used in our samples. We thank the reviewer for bringing this to our attention.

The revised manuscript reads well, and I am fully supportive of publication. Congratulations on an important piece of work for the tracking and MINFLUX community.

- Thank you very much for this very positive statement.

Response to the Reviewers

3rd Review

We thank all the reviewers for their time and commitment reading our manuscript in detail and for raising important comments once again. As in the previous revision, we tried to consider it and have revised the manuscript accordingly. Specifically, we have now left out the wording “iterative”.

Below we list the detailed responses to each point raised by the reviewers.

Reviewer I:

The authors have addressed most issues raised by the reviewers, and I can recommend its publication. Nonetheless, the paper is still not fully clear about the role of iterations to reduce L. In their rebuttal letter, the authors have explained why they focus on the later stage of the localization procedure, which is not iterating to reduce L. However, the manuscript is still misleading in parts, including (unnecessarily) the word "iterative" not only in the title but in all sub-titles and throughout the text. That said, I no longer wish to discuss this issue and leave it to the authors to consider clarifying this point in the final version of the paper.

We thank the referee for the constructive remark. To avoid confusion, we decided to omit the word *iterative* in correspondence to MINFLUX, thereby substituting phrases as follows:

- *iMFX* -> *MFX*
- Iterative MINFLUX -> MINFLUX